# Contextual effects in sensorimotor adaptation adhere to associative learning rules

Guy Avraham[1,2]*, Jordan A Taylor[3], Assaf Breska[1,2,4], Richard B Ivry[1,2], Samuel D McDougle[5]*

[1]Department of Psychology, University of California, Berkeley, Berkeley, United States; [2]Helen Wills Neuroscience Institute, University of California, Berkeley, Berkeley, United States; [3]Department of Psychology, Princeton University, Princeton, United States; [4]Max Planck Institute for Biological Cybernetics, Tübingen, Germany; [5]Department of Psychology, Yale University, New Haven, United States

**Abstract** Traditional associative learning tasks focus on the formation of associations between salient events and arbitrary stimuli that predict those events. This is exemplified in cerebellar-dependent delay eyeblink conditioning, where arbitrary cues such as a tone or light act as conditioned stimuli (CSs) that predict aversive sensations at the cornea (unconditioned stimulus [US]). Here, we ask if a similar framework could be applied to another type of cerebellar-dependent sensorimotor learning – sensorimotor adaptation. Models of sensorimotor adaptation posit that the introduction of an environmental perturbation results in an error signal that is used to update an internal model of a sensorimotor map for motor planning. Here, we take a step toward an integrative account of these two forms of cerebellar-dependent learning, examining the relevance of core concepts from associative learning for sensorimotor adaptation. Using a visuomotor adaptation reaching task, we paired movement-related feedback (US) with neutral auditory or visual contextual cues that served as CSs. Trial-by-trial changes in feedforward movement kinematics exhibited three key signatures of associative learning: differential conditioning, sensitivity to the CS-US interval, and compound conditioning. Moreover, after compound conditioning, a robust negative correlation was observed between responses to the two elemental CSs of the compound (i.e. overshadowing), consistent with the additivity principle posited by theories of associative learning. The existence of associative learning effects in sensorimotor adaptation provides a proof-of-concept for linking cerebellar-dependent learning paradigms within a common theoretical framework.

*For correspondence:
guyavraham@berkeley.edu (GA);
samuel.mcdougle@yale.edu
(SDMcD)

## Editor's evaluation

This paper provides a fundamental account of the role associative learning plays in sensorimotor adaptation. In a compelling result, the authors show that by pairing movement-related feedback with conditioning cues in the form of neutral auditory or visual contextual cues can be used to differentiate between sensorimotor perturbations/states. This work nicely integrates multiple literatures surrounding the processes supported by the cerebellum and solves a long-standing puzzle of exactly how and when arbitrary cues can serve to shape motor adaptation.

## Introduction

Sensorimotor adaptation refers to the gradual adjustment of movements in response to changes in the environment or body. In laboratory adaptation tasks, the introduction of perturbed sensory

feedback results in a sensorimotor prediction error. This error signal is used to update a model of an internal sensorimotor mapping, thus ensuring that the sensorimotor system remains well calibrated (*Shadmehr and Krakauer, 2008*; *Wolpert et al., 1995*; *Wolpert and Ghahramani, 2000*). The integrity of the cerebellum is essential for this form of error-based learning (*Donchin et al., 2012*; *Izawa et al., 2012*; *Popa et al., 2016*; *Schlerf et al., 2012*) and is thought to play a key role in generating predictions of future states given the current sensory context.

Different contexts may call for the use of different sensorimotor mappings. For instance, distinct motor memories should be retrieved when a skilled tennis player prepares to hit a service return on a clay court vs a grass court. A topic of considerable discussion in the adaptation literature centers on the constraints underlying effective contextual cues (*Addou et al., 2011*; *Addou et al., 2011*; *Heald et al., 2021*; *Dawidowicz et al., 2022*; *Heald et al., 2018*; *Howard et al., 2012*; *Howard et al., 2013*; *Howard et al., 2015*; *Karniel and Mussa-Ivaldi, 2002*; *Osu et al., 2004*; *Schween et al., 2019*; *Sheahan et al., 2016*; *Sheahan et al., 2018*; *Forano and Franklin, 2020*; *Forano et al., 2021*). Contextual cues are highly effective when the cue is directly relevant to properties of the movement. For example, people can simultaneously adapt to competing sensorimotor perturbations if the current context is established by the movement segment that precedes or follows the perturbed segment of a reach (*Howard et al., 2012*; *Howard et al., 2015*; *Sheahan et al., 2016*). In these situations, the contextual cues are hypothesized to be effective because the cues are incorporated into the motor plan (*Howard et al., 2012*; *Howard et al., 2013*).

Less clear is the efficacy of arbitrary contextual cues, ones that are not directly related to movement. In general, arbitrary cues do not appear to be effective for defining distinct motor memories. For example, participants show large interference between opposing perturbations that are signaled by color cues (*Howard et al., 2012*; *Howard et al., 2013*; *Gandolfo et al., 1996*; *Forano et al., 2021*). This interference can be reduced with multiple days of practice, although even after 10 days of training, residual interference remains on trials immediately following a switch in context (*Addou et al., 2011*; see also *Osu et al., 2004*).

The ineffectiveness of arbitrary contextual cues in shaping sensorimotor adaptation is surprising when one considers another popular model task of cerebellar-dependent sensorimotor learning – delay eyeblink conditioning. In this form of classical conditioning, a sensory cue (conditioned stimulus [CS], e.g. a tone) is repeatedly paired with an aversive event (unconditioned stimulus [US], e.g. an air puff to the cornea). By itself, the aversive event naturally causes an immediate unconditioned response (UR, e.g. an eye blink in response to the air puff). After just a short training period with a reliable CS, organisms as diverse as turtles and humans produce an adaptive conditioned response (CR, e.g. an eye blink that anticipates the aversive air puff). Notably, there is little constraint on the features of the predictive cues. The CR can be readily acquired in response to arbitrary CSs, such as a tone or a light flash. However, there is an important temporal constraint, at least with respect to cerebellar-dependent eyeblink conditioning. The cue must appear before the aversive event, and the two stimuli must be in close temporal proximity (*Schneiderman and Gormezano, 1964*; *Smith et al., 1969*).

In the present study, we set out to re-evaluate the efficacy of arbitrary cues in sensorimotor adaptation, drawing on key theoretical concepts and design features employed in studies of delay eyeblink conditioning. Using a variant of a visuomotor adaptation task, we show that arbitrary stimuli can act as effective contextual cues when a strict temporal relationship is established between those cues and their associated movement-related sensory outcomes. We then take this result a step further by showing that the effect of these arbitrary cues adheres to an established core principle of associative learning – cue additivity. The parallels between adaptation and conditioning observed in our results lay the empirical groundwork for a more parsimonious model of cerebellar-dependent sensorimotor learning.

## Results

Our primary goal in the present study was to determine if arbitrary stimuli can serve as effective contextual cues in sensorimotor adaptation. Participants reached from a start location to a target, with movement feedback provided by a cursor (*Figure 1A*). In typical adaptation tasks, the onset of the target serves as an imperative for movement initiation. When considered through the lens of classical conditioning, the target appearance could be viewed as a CS given that it is presented just before the sensory feedback (the US) resulting from the movement (*Figure 1B*). To test the efficacy of arbitrary

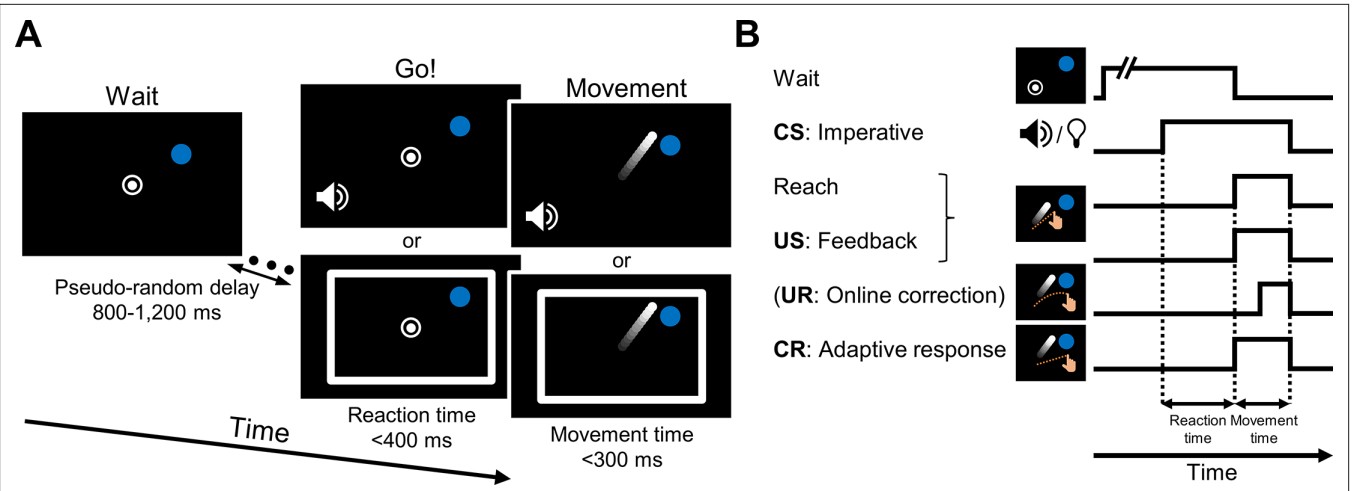

**Figure 1.** Task design. (**A**) Reaching task. Participants reach from a start location (white circle) to a target (blue dot). Online feedback is provided by a cursor (white dot). The target is displayed in a fixed location for the entire experiment (location varied across participants). After a random delay, a tone or a light (white frame) is presented, serving as the movement imperative. The participant is instructed to move directly to the target. The cue persists until the radial position of the hand reaches the target distance. The hand is not visible during the reach; instead, feedback is provided only by the cursor. (**B**) Components and timing of the reaching task described as conditioning events. The tone and light imperatives serve as conditioned stimuli (CSs), and the cursor feedback is the unconditioned stimulus (US). In our experiments, the radial position of the cursor is aligned with the hand, but the angular position is fixed ('clamped'), appearing either 15° away from the target (resulting in an error signal) when paired with one of the CS's (CS+) or at the target when paired with the other CS (CS−). After repeated pairings of the CS+ and US, a conditioned response (CR) develops, leading to an adaptive feedforward movement response in a direction opposite to the rotated feedback. Note that we include an unconditioned response (UR) in the schematic, what we assume is an online, automatic corrective response to the visual error. However, because participants are instructed to move fast, the UR is negligible.

cues as CSs, the onset of the tone or light served as the imperative, with the target visible at its fixed location throughout the experimental session (**Figure 1B**). For optimal delay eyeblink conditioning, the interval between the CS and the US is relatively narrow, on the order of 100–500 ms (**Schneiderman and Gormezano, 1964**; **Smith et al., 1969**). By using a neutral stimulus to cue movement initiation, we imposed a tight temporal link between the CS and the US, requiring the participant to initiate the movement within 400 ms of the onset of the imperative. Participants complied with this requirement, exhibiting rapid response times ([mean ± STD]: 293±49.0 ms).

A common perturbation technique to elicit sensorimotor adaptation is to introduce a visuomotor rotation, where the movement path of the visual feedback cursor is rotated (about the origin) with respect to the hand's path. This drives an adaptation process in which the movement heading angle shifts across trials in the direction opposite to the rotated cursor, thus offsetting the error (**Krakauer et al., 2000**). To continue with our conditioning analogy, the CR would be the movement heading angle elicited by a CS, expressed as a change in heading angle following the pairing of that CS with the visual error, the US (**Figure 1B**). We note here that the UR in an adaptation task could correspond to an immediate movement correction in response to the perturbation, that is, an 'online' correction. Given that we required fast movements to minimize these corrections, this potential form of UR is not observed in our design.

Rather than rotate the position of the feedback cursor with respect to the actual hand position, we opted to use a visual 'clamp' where the cursor followed an invariant path with respect to the target (**Morehead et al., 2017**; **Shmuelof et al., 2012**). In contrast to traditional movement-contingent feedback, the clamp method eliminates confounding effects that can come about from strategic processes (**McDougle et al., 2016**; **Kim et al., 2020**), and the size of the error can be controlled on every trial. The participant was fully informed of this manipulation and was instructed to ignore the task-irrelevant cursor and always reach straight to the target. Despite these instructions, participants' reach angles gradually shift in the direction opposite the error clamp, showing the key signatures of implicit sensorimotor adaptation (**Morehead et al., 2017**; **Kim et al., 2018**; **Tsay et al., 2020a**; **Tsay et al., 2021**; **Avraham et al., 2021**; **Poh et al., 2021**; **Vandevoorde and Orban de Xivry, 2019**; **Yin and Wei, 2020**).

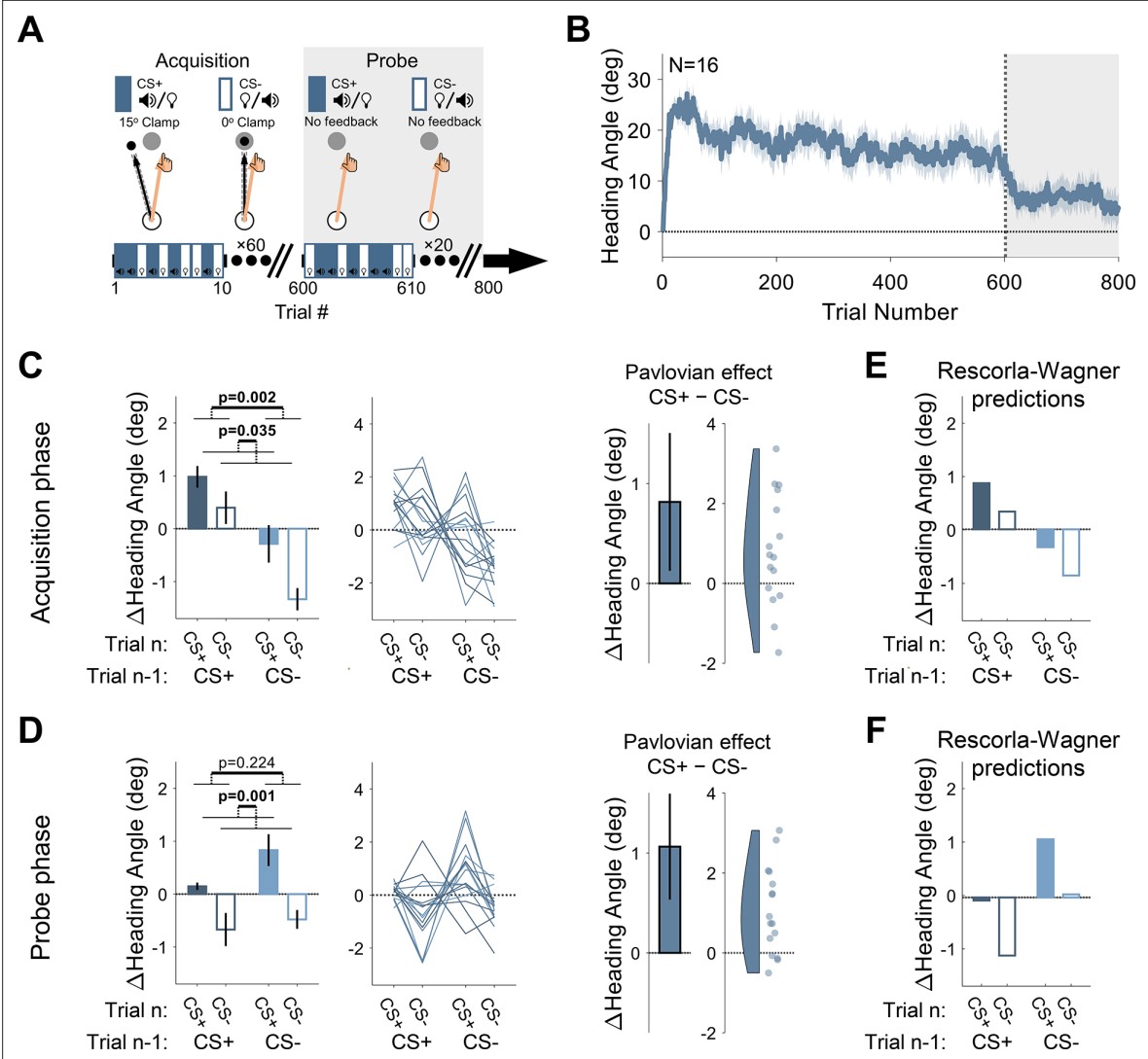

**Figure 2.** Experiment 1: differential conditioning. (**A**) Experimental protocol. During acquisition (white background), a 15° clamp (clockwise [CW]/counterclockwise [CCW], counterbalanced across participants) was associated with CS+ (e.g. a tone) and a 0° clamp with CS− (e.g. a light; counterbalancing the associations with the tone and light across participants). During the probe phase (gray background), the CS+ and CS− were presented without feedback. Throughout the entire experiment, CS+ and CS− trials were randomly interleaved. (**B**) Mean heading angle (N=16) as a function of trial number. Clamped feedback was presented on all trials during the acquisition phase (white background) and absent on all trials in the probe phase (gray background). Shaded region represents SEM. (**C and D**) Experimental results for trial-by-trial change (Δ) in heading angle (mean ± SEM) during the acquisition (**C**) and probe (**D**) phases. Left panels present the results of a two-way repeated-measures ANOVA for an adaptation effect (main effect of trial *n*−1, dark vs light blue) and a Pavlovian effect (main effect of the presented CS on the current trial *n*, filled vs empty bars). The black outlined bar and violin plot (right panel) present the Pavlovian effect, i.e., the subtraction of heading angle changes between CS+ and CS− trials (mean and 95% CI). (**E and F**) Rescorla-Wagner model simulation results during the acquisition (**E**), and probe (**F**) phases are consistent with the experimental results. Dots and thin lines represent individual participants. CS, conditioned stimulus.

The online version of this article includes the following source data for figure 2:

**Source data 1.** Related to *Figure 2B*.

**Source data 2.** Related to *Figure 2C*.

**Source data 3.** Related to *Figure 2D*.

## Sensorimotor adaptation is modulated by arbitrary sensory cues

In a standard differential conditioning design, one CS is paired with the US (CS+) and another CS is presented without the US (CS−). Thus, only the CS+ should become associated with the US and result in a CR. To implement this in Experiment 1, we used two arbitrary cues for the CSs – a tone or a light

cue (*Figures 1A*, *2A*). For the CS+ condition, the feedback cursor followed a clamped path that was rotated from the target by 15°. Rather than eliminate the US on CS− trials, we used a 0° clamp in which the feedback cursor always moved directly to the target. Thus, there was always a US on each trial, with the 15° US creating an error signal (paired with the CS+) and the 0° US signaling the absence of error (paired with the CS-).

During a 600-trial acquisition phase, CS+ and CS− trials were randomly interleaved. Participants exhibited a marked change in movement direction during this phase, reaching an asymptote of ~15° (*Figure 2B*). The observed rapid adaptation is consistent with previous adaptation studies, particularly those in which the target appears at a single fixed location (*Bond and Taylor, 2015*; *Day et al., 2016*; *McDougle et al., 2015*; *McDougle et al., 2017*; *Poh et al., 2021*).

The main analysis centered on trial-by-trial changes in heading angle. The change in heading angle from trial $n-1$ to trial $n$ is normally dictated by the feedback experienced in trial $n-1$. Thus, following experience with an error on CS+ trials, participants should show increased adaptation (a positive change in heading angle), and following no error on CS− trials, decreased adaptation (i.e. extinction). We refer to these trial-by-trial changes as the 'adaptation effect,' the standard measure of learning in sensorimotor adaptation tasks. There was a robust adaptation effect (*Figure 2C*, left panel). Trial-by-trial changes in reaching direction (Δ heading angle) were significantly affected by the CS presented on the previous trial (F[1, 15]=13.4, p=0.002, $BF_{10}$=41.45, and $\eta_p^2$=0.47) such that the change in heading angle was larger after CS+ trials compared to after CS− trials (mean difference: 1.50°, 95% CI: [0.70° 2.30°]). That is, the error occurring on CS+ trials resulted in learning that carried over to the next trial, whereas the absence of an error on a previous CS− trial resulted in a relative reversion to baseline (extinction). This is the canonical signature of incremental sensorimotor adaptation.

Importantly, the conditioning framework makes a critical prediction. The CS+ and CS− should differentially modulate the hand's heading angle on trial $n$ itself. That is, the presentation of the light or tone should produce a CR associated with that cue, leading to a difference in heading angle on trials with different cues that is independent of the feedback received on the previous trial. We refer to this as the 'Pavlovian effect.' As shown in *Figure 2C* (both panels), the results revealed clear Pavlovian effects during the acquisition phase. The heading angle increased in the direction of adaptation on CS+ trials (i.e. a positive change in heading angle relative to previous trials) and decreased on CS− trials (mean difference: 0.82°, 95% CI: [0.13° 1.51°]; F[1, 15]=5.37, p=0.035, $BF_{10}$=2.89, and $\eta_p^2$=0.26). This effect provides a novel demonstration that arbitrary sensory cues can lawfully influence implicit sensorimotor adaptation.

We also observed an interaction between CS identity on trials $n-1$ and $n$ (F[1, 15]=4.70, p=0.047, $BF_{10}$=2.21, and $\eta_p^2$=0.24). That is, the difference between CS+ and CS− was larger on trials following a CS− (mean: 1.05°, 95% CI: [0.20° 1.89°]) compared to trials following a CS+ (0.58°, [–0.13° 1.30°]). This interaction effect may reflect an asymmetry between the rate of the acquisition and extinction processes once CS-US associations are established. That is, the state following a CS+ trial may be closer to its asymptotic limit than after a CS− trial and is thus more limited in its potential for further change.

We note that the visual feedback was different on CS+ and CS− trials, with the cursor deviating from the target in the former and moving in a straight line to the target in the latter. This raises the possibility that the heading angle differences on CS+ and CS− trials could be affected by online feedback corrections. This explanation is unlikely given that the movements were quite rapid (mean movement time [MT] ± STD, 103±31.2 ms). To assess the online correction hypothesis more directly, we calculated the difference in heading angle 50 ms after movement initiation and at the radial distance of the target. There was no overall change in heading angle between the time points (–0.32°±2.92°) and no significant difference between CS+ and CS− trials on this metric (mean difference: –0.04°, 95% CI: [–0.19° 0.12°]; t[15]=–0.47, p=0.640, $BF_{10}$=0.28, and d=–0.12). Thus, our results appear to pertain exclusively to feedforward learning.

Following the acquisition phase, participants completed a probe phase in which CS+ and CS− trials were randomly presented in the absence of any visual feedback (no US). This phase provides a clean test for associative learning effects since it removes trial-by-trial effects that arise from the differential feedback given during CS+ and CS− acquisition trials. Here too, we observed a clear Pavlovian effect. Although there was an overall decrease in heading angle across the probe phase (i.e. a partial washout of adaptation, *Figure 2B*), there was a significant main effect of the CS presented on trial $n$ (1.06°,

[0.53° 1.59°]; F[1, 15]=15.45, p=0.001, BF$_{10}$=72.15, and $\eta_p^2$=0.51; *Figure 2D*), with a relative increase in heading angle on CS+ trials and a decrease on CS− trials.

Despite the absence of feedback in the probe phase, the change in heading angle on trial *n* (CS+ or CS−) should also depend on the CS presented on trial *n*−1. After an association is formed, the motor state, namely the actual heading angle, fluctuates as a function of the presented CS, decreasing on CS− trials (due to extinction) and increasing on CS+ trials (due to conditioning). As such, the heading angle on a CS+ trial should increase more if it follows a CS− trial than if it follows a CS+ trial. Similarly, the heading angle on a CS− trial should decrease more if it follows a CS+ trial than if it follows a CS− trial. While the data appeared to follow this pattern, there was no main effect of the trial *n*−1 CS (−0.44°, [−1.11° 0.24°]; F[1, 15]=1.61, p=0.224, BF$_{10}$=0.56, and $\eta_p^2$=0.10). This may be because, statistically, the main effect of trial *n*−1 considers consecutive trials with a repeated CS (CS+ that follows CS+, and CS− that follows CS−) where the change in heading angle is minimal. Crucially, the Pavlovian effect (the difference between the change in heading angle on CS+ and CS− trials on trial *n*) should not depend on the previous trial. Consistent with this prediction, the trial *n*−1 CS × trial *n* CS interaction was not significant (F[1, 15]=2.03, p=0.175, BF$_{10}$=0.69, and $\eta_p^2$=0.12).

The Pavlovian effect did not appear to be driven by explicit awareness of the CS-US contingency. A post-experiment survey was used to classify participants as either aware or unaware of the CS-US associations (see Materials and methods). Participants who reported being aware of the contingencies (N=7) did not show a different Pavlovian effect compared to those who reported being unaware (N=9) in either the acquisition (−0.34°, [−1.90° 1.22°]; t[14]=−0.470, p=0.647, BF$_{10}$=0.46, and d=−0.24) or probe (−0.14°, [−1.34° 1.07°]; t[14]=−0.242, p=0.813, BF$_{10}$=0.44, and d=−0.12) phases. A similar null effect of awareness on the strength of conditioning has also been reported in studies of human delay eyeblink conditioning (*Clark and Squire, 1998*).

In summary, the observed effects of context on implicit sensorimotor adaptation in both the acquisition and probe phases in Experiment 1 are consistent with differential conditioning effects. Feedforward implicit sensorimotor adaptation – here operationalized as a type of CR – was differentially modulated by arbitrary sensory CSs, with a greater response to the CS+, the cue that was paired with a visuomotor error.

## The Rescorla-Wagner model for context-dependent sensorimotor adaptation

One influential model that has been used to describe contextual effects in associative learning tasks is the Rescorla-Wagner model (*Rescorla and Wagner, 1972*). This model formalizes changes in CRs via the modulation of learned associations. Here, the associative strengths, *V*, of the conditioning stimuli are updated according to the learning rule (*Equation 1*):

$$V^{[n]} = V^{[n-1]} + \alpha \cdot \beta \cdot SPE^{[n-1]}; \ SPE^{[n-1]} = \lambda - V^{[n-1]} \tag{1}$$

where *V* represents the associative strengths between the US and the CS. It is updated based on the sensory prediction error (SPE) presented on trial *n*−1. The SPE is defined as the difference between the maximum conditioning (asymptotic) level for the US (*λ*) and the associative strength on the given trial. *β* is the learning rate parameter of the US, and *α* represents the salience of the CS. We note that the Rescorla-Wagner model is one of many computational frameworks for associative learning (*Courville et al., 2006*; *Gershman, 2015*) and does not provide a mechanistic account for the error-correction process itself (e.g. the fact that the motor system 'knows' to update movements in the direction opposite of the error). For simplicity, we assume that the sign of the change in movement direction is coded in specialized neural mechanisms for reducing motor error (*Hadjiosif et al., 2021*; *Herzfeld et al., 2018*; *Wolpert et al., 1998*).

To examine whether the Rescorla-Wagner model can capture differential conditioning behavior similar to the results observed in Experiment 1, we performed model simulations. The simulation results demonstrated both the adaptation effect and Pavlovian effect for the acquisition phase (*Figure 2E*), as well as a clear Pavlovian effect for the probe phase (*Figure 2F*). Importantly, the behavioral signature of differential conditioning – the larger changes in heading angle on CS+ trials compared to CS− trials – holds for essentially all combinations of parameters in the Rescorla-Wagner model (*Figure 3*, left side).

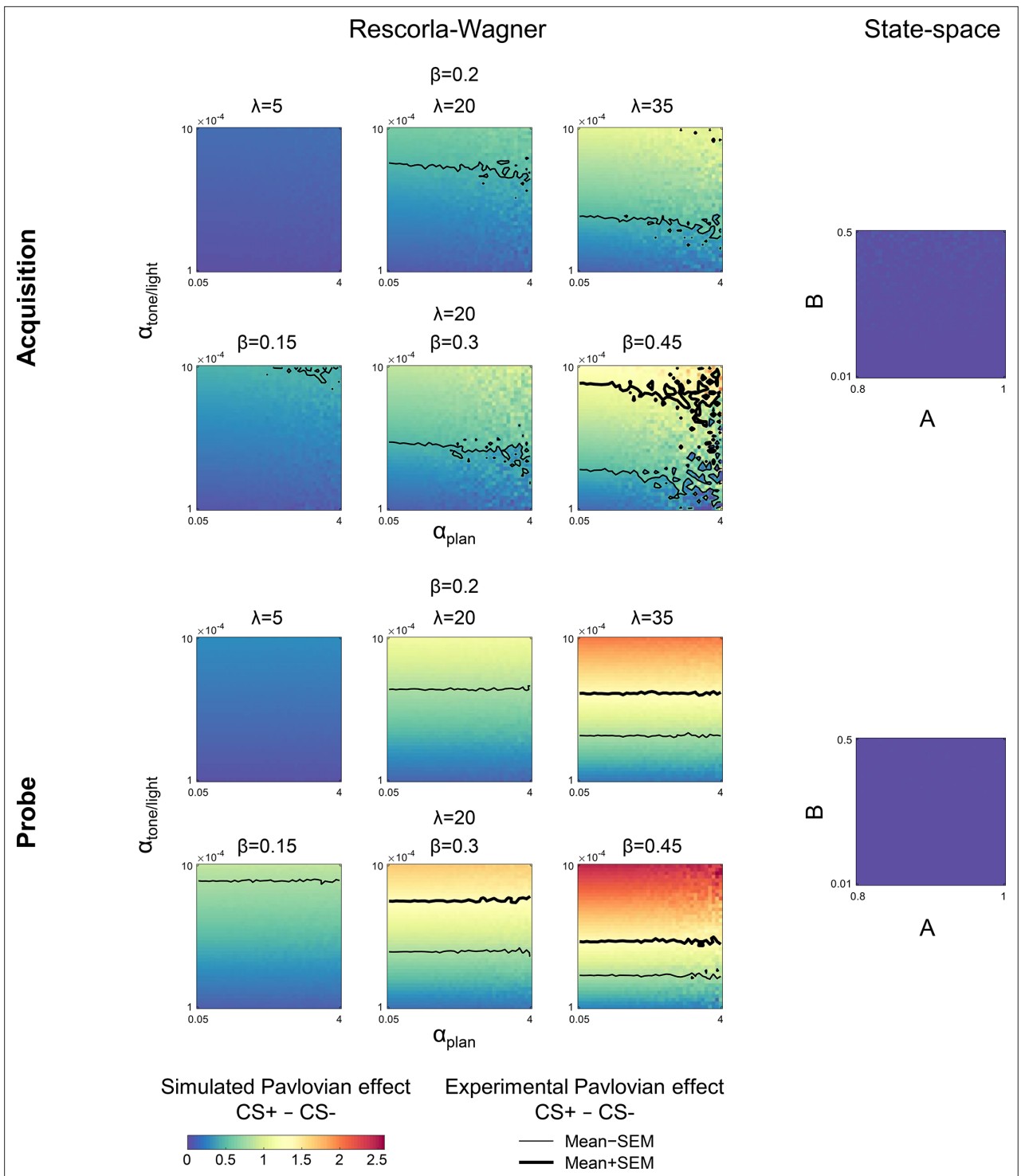

**Figure 3.** Simulations of the Pavlovian effect due to differential conditioning in Experiment 1. Simulations of the current trial Pavlovian effect, the difference between the change in heading angle on CS+ and CS− trials, for different parameter values as predicted by the Rescorla-Wagner model (left side). The results for the acquisition phase are shown on top and probe phase on bottom. Contours represent the mean ± SEM of the Pavlovian effect from Experiment 1. All parameter values for the state-space model fail to predict a differential response on the current trial for CS+ and CS− trials (right side). CS, conditioned stimulus.

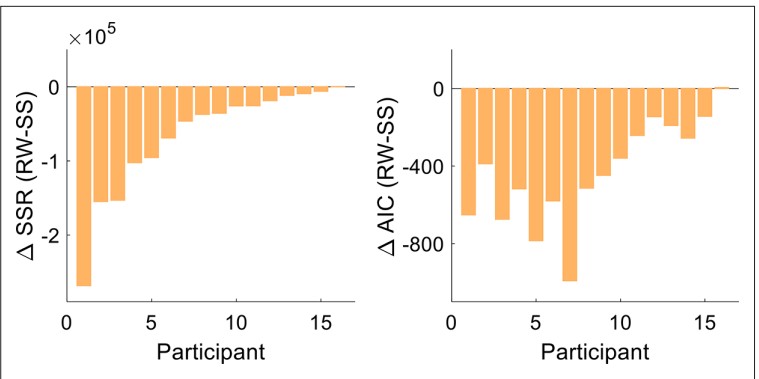

**Figure 4.** Model comparison. The sum of squares residuals (SSR) and Akaike information criterion (AIC) difference between the Rescorla-Wagner (RW) and the state-space (SS) models for each participant of Experiment 1.

Sensorimotor adaptation is typically described by a 'state-space' model in which the motor state (*x*) is updated according to the following learning rule (*Equation 2*):

$$x^{[n]} = A \cdot x^{[n-1]} + B \cdot SPE^{[n-1]} \tag{2}$$

where SPE is the sensory prediction error – the difference between the predicted and the actual sensory feedback – experienced on trial *n*−1, *A* is the retention factor, and *B* is the learning rate. The state-space model is broadly similar to the Rescorla-Wagner (e.g. both models share the Markov property, produce exponential-family learning curves, etc.). However, the basic state-space model is unable to account for context effects. Unlike the Rescorla-Wagner model, it does not include parameters that allow the updating of separate states that are associated with distinct contexts. It predicts the change in behavior based on the outcome of the previous trial (i.e. the adaptation effect); it cannot account for variation in heading angle due to the CS on the current trial (i.e. the Pavlovian effect; *Figure 3*, right side).

To formally compare the Rescorla-Wagner and state-space models in terms of the observed results, we conducted a model comparison by fitting each participant's time course data with the two models. The Rescorla-Wagner model provided a better fit to the data than a standard state-space model, underscoring the significant role of the Pavlovian effect (*Figure 4*; *t* test comparing sum of squared

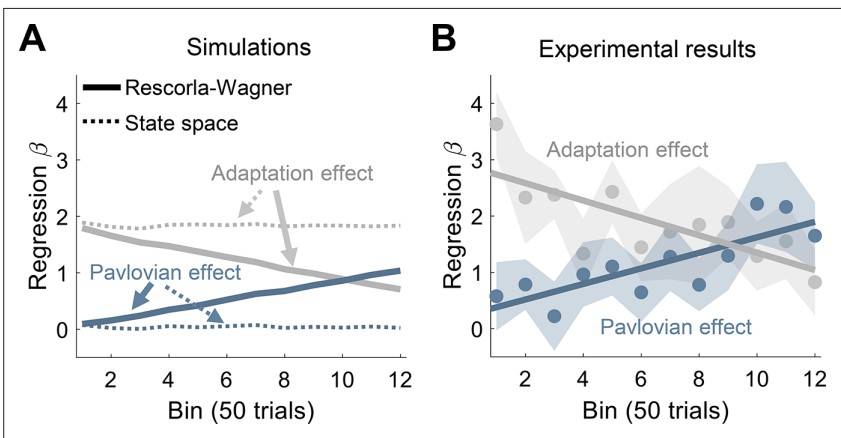

**Figure 5.** Dynamics of adaptation and Pavlovian effects. (**A**) Time course of the mean weights (least squares regression *β*) of the adaptation (trial *n*−1, gray) and Pavlovian (trial *n*, blue) effects in Experiment 1 as predicted by the Rescorla-Wagner model (solid lines) and the state-space model (dotted lines). (**B**) Time course of the mean weights of the adaptation and Pavlovian effects as derived from fits of the experimental results. Shaded regions represent SEM.

The online version of this article includes the following source data for figure 5:

**Source data 1.** Related to *Figure 5B*.

residuals, $-6.61 \times 10^4$, [$-1.05 \times 10^5$ $-2.72 \times 10^4$]; t[15]=$-3.62$, p=0.003, and d=$-0.90$; t test comparing Akaike information criterion (AIC) values, $-430$, [$-572$ $-288$]; t[15]=$-6.44$, p<0.001, and d=$-1.61$).

An additional analysis provided further support for an associative learning account of the results of Experiment 1. The Rescorla-Wagner model not only provides a framework for understanding how associations can be formed with arbitrary stimuli, but it can also capture how the strength of these associations is constrained by the relevance of the cues. For instance, gustatory cues are much more likely to be associated with an internal bodily state (e.g. nausea) than a visual cue (*Garcia and Koelling, 1966*). In the current study, the clamped feedback (the US) is a highly relevant stimulus for reaching; as such, we should expect it to have an immediate strong influence on motor behavior. In contrast, the imperative cues (the tone and light CSs) have no natural relevance for reaching; as such, their contribution to the CR should initially be quite modest but gradually increase with experience (*Figure 5A*). To test this prediction, we examined the time course of the adaptation and Pavlovian effects during acquisition using a linear mixed model analysis. As expected, at early stages of acquisition, the adaptation effect emerged rapidly, whereas the contribution of the Pavlovian effect was small. Over experience, the relative contribution of the two effects reversed (type X time bin interaction effect: F[1, 365]=16.7, p<0.001). The Pavlovian effect gradually grew (mean slope: 0.14, 95% CI: [0.04 0.24]) at the expense of the adaptation effect, which eventually exhibited a reduced contribution ($-0.15$, [$-0.25$ $-0.05$]; *Figure 5B*). We note that both the decrease in the adaptation effect and the increase in the Pavlovian effect are not captured by a typical state-space model (*Figure 5A*).

## Pavlovian effects in sensorimotor adaptation are sensitive to the CS-US interval

The Pavlovian effects observed in Experiment 1 provide evidence that sensorimotor adaptation can be shaped by arbitrary contextual cues. These results stand in contrast to previous work showing that arbitrary cues are ineffective for separating competing motor memories (e.g. differential responses to clockwise [CW] and counterclockwise [CCW] rotations). This null result has been taken to reflect the irrelevance of such cues to the motor state (*Gandolfo et al., 1996*; *Karniel and Mussa-Ivaldi, 2002*; *Howard et al., 2012*; *Howard et al., 2013*).

Why were the arbitrary cues effective in Experiment 1? Inspired by studies of eyeblink conditioning, we opted to impose a strong temporal constraint on the CS-US interval. The contextual cues served as the imperative signals, and we required that the participants respond quickly to the imperative, resulting in a mean CS-US interval of around 300 ms. The timing of events was somewhat more relaxed in prior studies. For example, in *Howard et al., 2013*, the color cues (which could be seen as CSs for differential conditioning) were presented 1000 ms before the imperative. When adding in the reaction time (RT), the CS-US interval may lie beyond the optimal value for cerebellar-dependent learning (*Brudner et al., 2016*; *Kitazawa et al., 1995*; *Schneiderman and Gormezano, 1964*; *Smith et al., 1969*). Thus, the reported ineffectiveness of arbitrary cues in separating motor memories may be related to the failure to meet the temporal requirements for forming associations between these cues and their associated feedback signals.

In Experiment 2, we tested the hypothesis that extending the temporal interval between arbitrary cues and sensorimotor feedback would attenuate or abolish the Pavlovian effect in visuomotor adaptation. To extend the interval, we opted to have the CSs no longer serve as the movement imperative; rather, a new cue, a color change of the target, served as the imperative signal (*Figure 6A*). To extend the CS-US interval, we imposed a delay between the onsets of the CS and the imperative. The mean of the delay was set to 1000 ms, with the actual value selected from a distribution ranging from 800 to 1200 ms to maintain unpredictability of the imperative timing (see Materials and methods for details).

During the acquisition phase of Experiment 2, participants exhibited a marked change in movement direction, opposite to the direction of the clamped feedback (adaptation effect: F[1, 63]=47.3, p<0.001, $BF_{10}$=$7.58 \times 10^6$, $\eta_p^2$=0.43, mean difference, 95% CI; 1.80°, [1.29° 2.31°]), reaching a mean asymptote of ~25° (*Figure 6B*). The asymptotic values in this experiment were higher than those seen in Experiment 1, possibly due to differences in the experimental setup (see Discussion). Despite this increase in overall adaptation, a Pavlovian differential conditioning effect was not detected (F[1, 63]=0.54, p=0.465, $BF_{10}$=0.16, $\eta_p^2$=0.01, mean difference: 0.17°, 95% CI: [$-0.29°$ 0.63°]), and there was no trial $n-1 \times n$ interaction (F[1, 63]=0.60, p=0.442, $BF_{10}$=0.17, and $\eta_p^2$=0.01; *Figure 6C*). During the probe phase, the trial $n-1$ effect disappeared with the removal of the visual feedback (F[1,

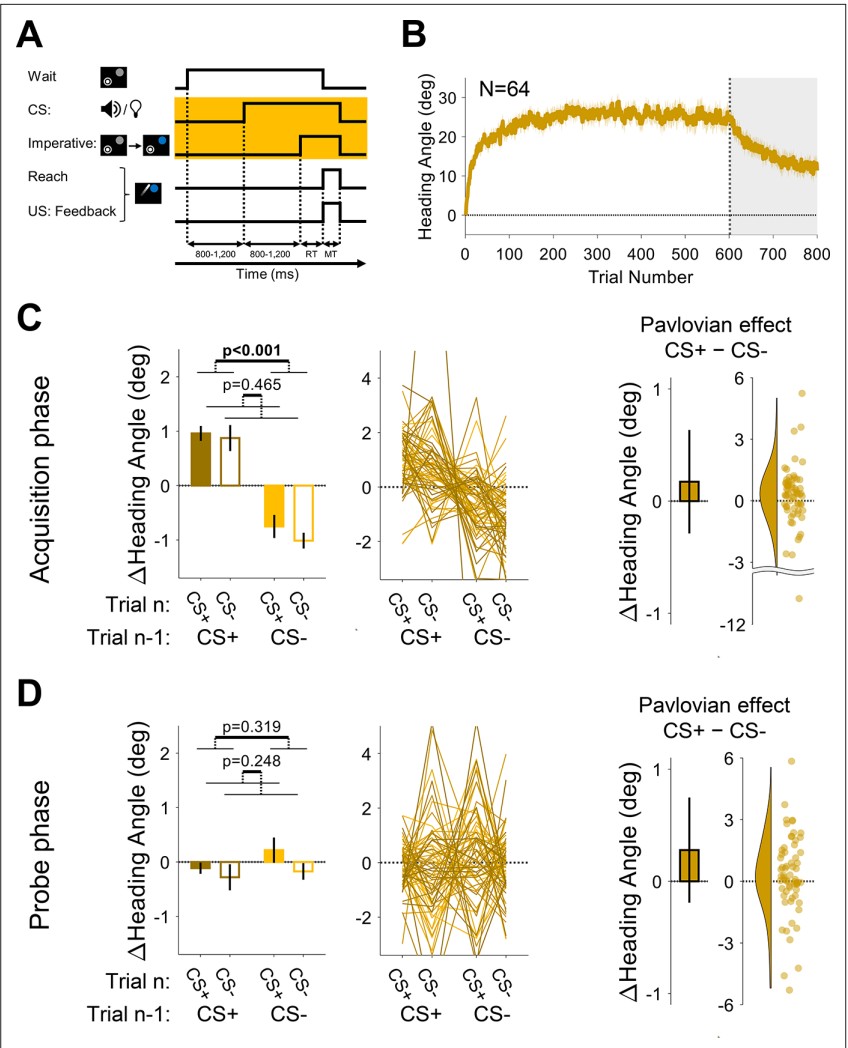

**Figure 6.** Experiment 2: Pavlovian effect in differential conditioning is abolished when the delay between the conditioned stimulus (CS) and unconditioned stimulus (US) is long. (**A**) Illustration of trial events and their timing. By using a separate stimulus for the CS (tone or light) and the imperative (target changed from gray to blue), a delay of ~1000 was imposed between the CS and US. The yellow background highlights the design difference from Experiment 1 (and Experiment 3 below). (**B**) Mean heading angle (N=64) as a function of trial number. Shaded region represents SEM. (**C and D**) Trial-by-trial change in heading angle (mean ± SEM) during the acquisition (**C**) and probe (**D**) phases. Left panels present the results of a two-way repeated-measures ANOVA for an adaptation effect (main effect of trial *n*−1, dark vs light colors) and a Pavlovian effect (main effect of the presented CS on the current trial *n*, filled vs empty bars). The black outlined bars and violin plots (right panel) present the Pavlovian effect, i.e., the subtraction of heading angle changes between CS+ and CS− trials (mean and 95% CI). Dots and thin lines represent individual participants.

The online version of this article includes the following source data for figure 6:

**Source data 1.** Related to *Figure 6B*.

**Source data 2.** Related to *Figure 6C*.

**Source data 3.** Related to *Figure 6D*.

63]=1.01, p=0.319, $BF_{10}$=0.21, $\eta_p^2$=0.02, mean difference: –0.22°, 95% CI: [–0.65° 0.21°]). Importantly, and consistent with the results from the acquisition phase, we again found no reliable Pavlovian effect (F[1, 63]=1.36, p=0.248, $BF_{10}$=0.25, $\eta_p^2$=0.02, mean difference: 0.28°, 95% CI: [–0.19° 0.75°]) and no trial *n*−1 × *n* interaction (F[1, 63]=0.42, p=0.517, $BF_{10}$=0.16, and $\eta_p^2$=0.01; *Figure 6D*). These results suggest that extending the CS-US interval by approximately 1000 ms rendered arbitrary sensory contextual cues ineffective in separating distinct motor memories.

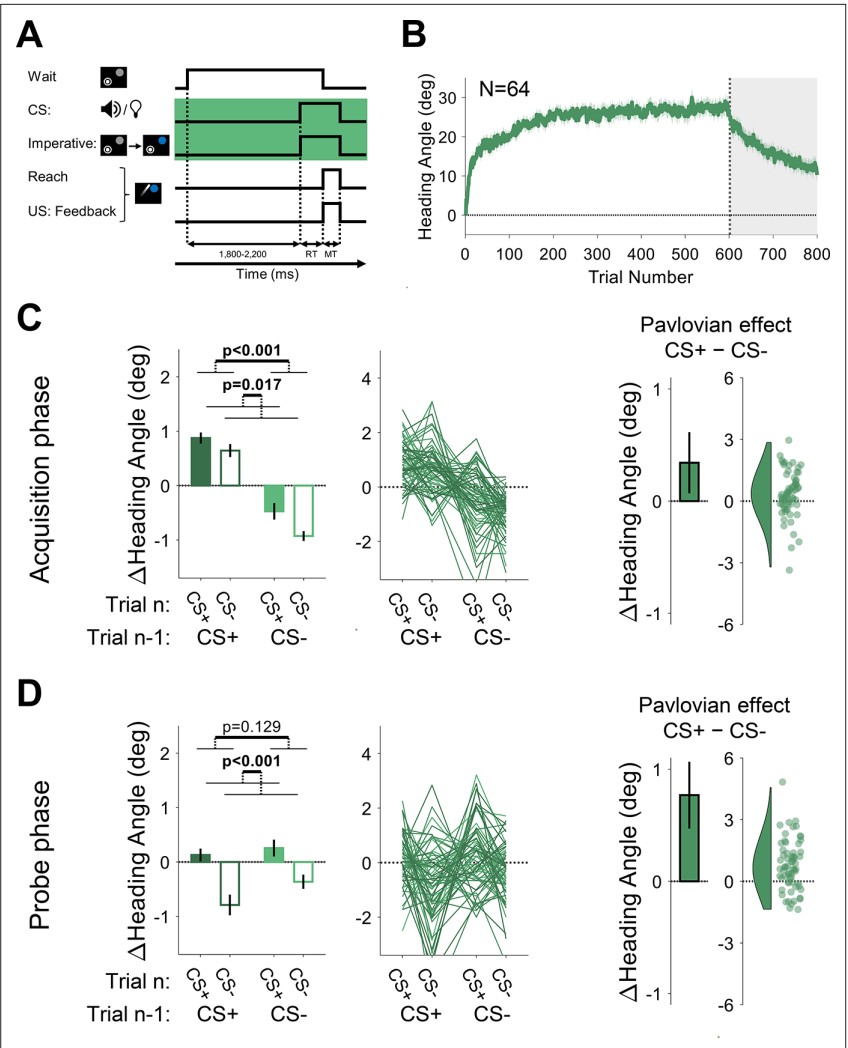

**Figure 7.** Experiments 3: Pavlovian effect in differential conditioning is observed when the conditioned stimulus (CS) and imperative cue occur simultaneously. (**A**) Illustration of trial events and their timing. The green background highlights the design difference from Experiment 2, namely the simultaneous presentation of the CS and the movement imperative. (**B**) Mean heading angle (N=64) as a function of trial number. Shaded region represents SEM. (**C and D**) Trial-by-trial change in heading angle (mean ± SEM) during the acquisition (**C**) and probe (**D**) phases. Left panels present the results of a two-way repeated-measures ANOVA for an adaptation effect (main effect of trial *n*−1, dark vs light colors) and a Pavlovian effect (main effect of the presented CS on the current trial *n*, filled vs empty bars). The black outlined bars and violin plots (right panel) present the Pavlovian effect, i.e., the subtraction of heading angle changes between CS+ and CS− trials (mean and 95% CI). Shaded regions and error bars represent SEM. Dots and thin lines represent individual participants.

The online version of this article includes the following source data for figure 7:

**Source data 1.** Related to *Figure 7B*.

**Source data 2.** Related to *Figure 7C*.

**Source data 3.** Related to *Figure 7D*.

One possible concern with our method of increasing the CS-US interval is that, by using a different stimulus as the imperative (the target color change), the salience of the tone and light CSs may have been reduced since the task no longer required the participants to attend to these cues. To address this, in Experiment 3, we used the same CSs (tone and light) and imperative signal (color change of the target) as in Experiment 2 but now made the onsets of the CS and imperative simultaneous on each trial (*Figure 7A*). In this way, the CS-US interval is much shorter [mean reaction time (RT) ± STD]: 306±57.9 ms), similar to Experiment 1. We also increased the inter-trial interval in Experiment

3 to roughly match that of Experiment 2 given evidence showing that the rate of associative events impacts classical conditioning (*Gallistel and Gibbon, 2000*).

The time course of adaptation in Experiment 3 was similar to that observed in Experiment 2, with a mean asymptote around ~25° (*Figure 7B*). During the acquisition phase, trial-by-trial changes in heading angle showed the standard adaptation (trial $n-1$) effect (F[1, 63]=101.6, p<0.001, BF$_{10}$=2.77×10$^{12}$, $\eta_p^2$=0.62, mean difference, 95% CI; 1.46°, [1.17° 1.74°]). However, unlike Experiment 2, we now observed a significant Pavlovian effect (F[1, 63]=6.05, p=0.017, BF$_{10}$=2.35, $\eta_p^2$=0.09, mean difference, 95% CI; 0.34°, [0.07° 0.61°]; *Figure 7C*). Consistent with the Rescorla-Wanger predictions (see *Figure 2E*), there was no significant interaction between trial $n-1$ and trial $n$ effects (F[1, 63]=1.30, p=0.258, BF$_{10}$=0.24, and $\eta_p^2$=0.26). The Pavlovian effect was also significant in the probe phase (F[1, 63]=25.6, p<0.001, BF$_{10}$=6.88×10$^3$, $\eta_p^2$=0.29, mean difference, 95% CI; 0.77°, [0.47° 1.07°]) with neither trial $n-1$ (F[1, 63]=2.37, p=0.129, BF$_{10}$=0.41, $\eta_p^2$=0.04, mean difference, 95% CI; –0.28°, [–0.63° 0.08°]) nor interaction effects (F[1, 63]=0.86, p=0.358, BF$_{10}$=0.19, and $\eta_p^2$=0.01; (*Figure 7D*)).

Taken together, the results of Experiments 2 and 3 support our conjecture that the efficacy of arbitrary stimuli in serving as contextual cues for sensorimotor conditioning is subject to strong temporal constraints, a key feature of cerebellar-dependent learning. Even when the tone and light no longer required attention, they proved effective for differential conditioning when the interval between these CSs and the US was short (around 300 ms) but not when the interval was extended (around 1000 ms).

## Additivity principle in response to compound stimuli is observed in sensorimotor adaptation

The results of Experiments 1–3 demonstrate that implicit sensorimotor adaptation displays two prominent features of associative learning – the associability of sensorimotor feedback with arbitrary sensory cues, and the key role of CS-US timing in the formation of those associations. In Experiment 4, we tested another core feature of associative learning, the principle of additivity (*Mackintosh, 1976*; *Pavlov, 1927*; *Rescorla and Wagner, 1972*). This principle is based on the idea that there is an associative capacity for a given US – the *V* term in the Rescorla-Wagner equation. That is, multiple CSs can become associated with a given US, but the combined associative strength is bounded by *V*. As a result of this capacity constraint, CSs effectively compete with one another, with the associative strength split among multiple cues.

The classic method to examine additivity is compound conditioning, where two or more stimuli are presented simultaneously to form a 'compound' CS (*Equation 3*). When paired with a US, this compound CS will come to elicit CRs. Importantly, the associative strength of the compound CS ($V_{comp}$) is the sum of the associative strengths of the elemental CSs ($V_i$), where $nS$ in *Equation 3* represents the number of elements forming the compound. Consequently, each element of the compound, when presented alone, elicits a proportionally weaker CR, with the degree of attenuation being a function of the associative strength of that CS.

$$V_i^{[n]} = V_i^{[n-1]} + \alpha \cdot \beta \cdot (\lambda - V_{comp}^{[n-1]}); \quad V_{comp}^{[n-1]} = \sum_{i=1}^{nS} V_i^{[n-1]} \qquad (3)$$

The additivity principle has received ample support in behavioral and neural studies of associative learning (*Giurfa, 2007*; *Kehoe et al., 1994*; *Kehoe and Schreurs, 1986*; *Rescorla and Wagner, 1972*; *Weiss, 1972*) but has not, to our knowledge, been tested in sensorimotor adaptation. In Experiment 4, we used a compound conditioning design, pairing a 15° error clamp stimulus with a compound CS (simultaneous presentation of the tone and light; *Figure 8A*) on all trials of the acquisition phase. As in Experiments 1–3, we again observed robust adaptation in the acquisition phase, manifesting as a change in heading angle in the direction opposite the clamp (*Figure 8B*).

The critical test in this experiment comes from the probe phase. Here, the visual feedback was eliminated, and the presented imperative was either the original compound CS or just the tone or light alone. We observed a significant Pavlovian effect of CS type on these no-feedback trials (Friedman test: $\chi^2$[2]=9.82, p=0.007, and W=0.223), with larger changes in heading angles on compound CS trials (median: 0.39°, interquartile range: [–0.05° 0.49°]) relative to the tone-alone (–0.15°, [–0.47° 0.18°], p=0.020) or light-alone (–0.22°, [–0.52° 0.34°]p=0.020) trials, conforming to the first prediction of the additivity principle (*Figure 8C and D*).

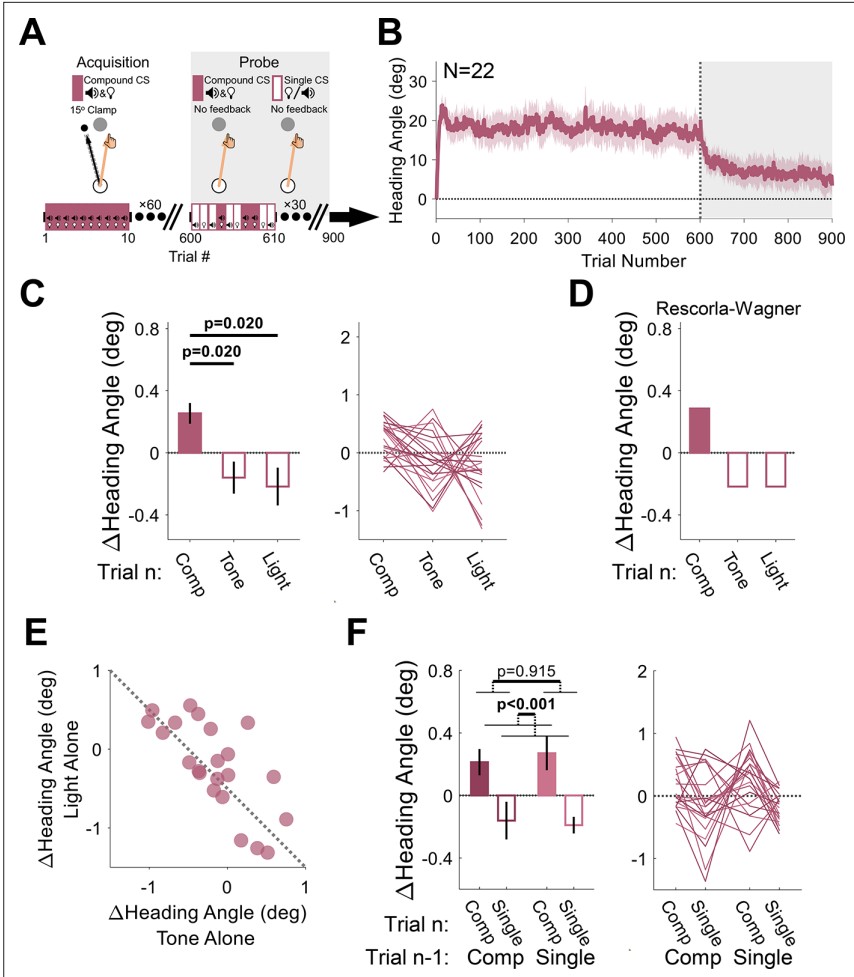

**Figure 8.** Experiment 4: compound conditioning. (**A**) During acquisition, a tone and light were presented simultaneously (compound conditioned stimulus [CS]), serving as the imperative signal for the reaching movement. They were paired with a 15° clamp on all acquisition trials (white background). During the probe phase (gray background), no feedback was provided, and the CSs were presented either together (compound CS) or alone (single CS; tone or light). (**B**) Mean heading angle (N=22) as a function of trial number. Shaded region represents SEM. (**C**) Experimental results for the Δ heading angle (median ± SEM) during the probe phase, showing a positive relative change for the compound CS on the current trial $n$ (filled bar) and a negative relative change for each of its elements (empty bars). Friedman test: $\chi^2[2]=9.82$, p=0.007. (**D**) Predictions from the Rescorla-Wagner model for trial-by-trial Δ heading angle during the probe phase. The two elements were assumed to have equal weight in the simulation. (**E**) Scatter plot showing the between-participant trade-off in terms of the associative strength of the two CSs (the dotted black line represents the unity line). (**F**) The Δ heading angle (mean ± SEM) after pooling the two single-CS conditions, and the results of a two-way repeated-measures ANOVA for the effects of the previous CS type (compound vs singleton, dark and light pink, respectively) and current CS (filled and empty bars). Thin lines represent individual participants.

The online version of this article includes the following source data for figure 8:

**Source data 1.** Related to *Figure 8B*.

**Source data 2.** Related to *Figure 8C and E*.

**Source data 3.** Related to *Figure 8F*.

We note that the median trial-by-trial change in heading angle for the tone- and light-alone trials is negative. This is because this measure reflects modulations in behavior around the mean heading angle, and this mean decreases over time due to extinction given the absence of the error feedback. This general trend should also vary for the different CSs depending on their relative salience (*Equation 3*; *Kamin, 1967*). Crucially, the additivity principle posits that there should be a negative

correlation between the associative strengths of competing CSs (*Rescorla and Wagner, 1972*). That is, if a strong associative bond is formed between one CS and the US, this will come at the expense of the associative strength accrued by competing CSs (*Equation 3*) given the capacity limit on associability (*V*). This prediction was strikingly confirmed in an analysis of the heading angle changes on tone- and light-alone trials. Participants who were more sensitive to the tone stimulus were less sensitive to the light stimulus and vice versa (*Figure 8E*, Pearson correlation: $r=-0.72$, $p<0.001$, and $BF_{10}=163.2$).

As in Experiments 1–3, since the probe phase of Experiment 4 consisted of different types of CSs presented randomly across trials, the behavior for a given trial should reflect not only the CS on trial $n$ but also the motor state on trial $n-1$ (which is itself also influenced by the CS on that trial). As a further test of compound conditioning, we pooled the two single-CS conditions to measure the effects of the previous and current CS type (singleton vs compound) on the observed changes in heading angle during the probe phase. We again observed the Pavlovian effect (mean: 0.42°, 95% CI: [0.24° 0.59°]; $F[1, 21]=21.2$, $p<0.001$, $BF_{10}=459.9$, and $\eta_p^2=0.50$). There was a relative increase in heading angle for the compound CS (0.24°, [0.13° 0.35°]) and a relative decrease for the singleton CSs (–0.18°, [–0.28° –0.07°]; *Figure 8F*). The main effect of the CS on trial $n-1$ was neither significant (–0.01°, [–0.27° 0.24°]; $F[1, 21]=0.012$, $p=0.915$, $BF_{10}=0.21$, and $\eta_p^2=5.58\times10^{-4}$) nor was the trial $n-1$ × trial $n$ interaction ($F[1, 21]=0.276$, $p=0.605$, $BF_{10}=0.25$, and $\eta_p^2=0.01$), presumably due to the elimination of adaptation effects given the absence of visual feedback in the probe phase.

## Discussion

Sensorimotor adaptation and delay eyeblink conditioning are foundational paradigms for studying error-based sensorimotor learning. They have yielded a rich empirical foundation in the development of theoretical and neural models of learning and memory, particularly with respect to cerebellar function (*Albus, 1971*; *Ito, 1984*; *Marr, 1969*; *Wolpert et al., 1998*). Adaptation has typically been modeled from a control engineering perspective, centered on the idea that changes in behavior are driven by sensory prediction errors that arise when experienced feedback deviates from that predicted by a forward model operating on an efference copy of the motor command (*Krakauer et al., 2019*; *Shadmehr and Krakauer, 2008*; *Wolpert and Flanagan, 2001*). In contrast, eyeblink conditioning is treated as an associative learning process that can be shaped by arbitrary sensory context cues.

The present study takes a first step toward a more parsimonious framework that might account for these two canonical forms of cerebellar-dependent motor learning. We were motivated by the apparent paradox concerning whether arbitrary stimuli can act as sufficient contextual cues for the establishment of distinct motor memories during sensorimotor adaptation. In attempting to resolve this paradox, we conceptualized adaptation as analogous to associative learning. Seen from this perspective, we recognized that prior studies failing to observe adaptation in response to arbitrary context cues had not considered the short, precise temporal relationship between these cues and sensorimotor feedback. By modifying a visuomotor adaptation task to conform to timing constraints similar to those required for delay eyeblink conditioning, we showed that adaptation exhibited the hallmarks of both differential conditioning (Experiments 1 and 3) and compound conditioning (Experiment 4). These results provide the first evidence, to our knowledge, that pairing neutral stimuli (i.e. tones and lights) with distinct visuomotor outcomes can differentially influence implicit feedforward sensorimotor adaptation in a manner consistent with core principles of associative learning.

The role of context in sensorimotor adaptation has been the subject of considerable debate, with a large body of empirical studies asking whether different sensory cues are sufficient to 'tag' different motor memories. Postural and movement-related variables, such as different lead-in and follow-through movements, are very effective contextual cues and allow participants to respond to interleaved, opposing perturbations with minimal interference (*Howard et al., 2012*; *Howard et al., 2013*; *Howard et al., 2015*; *Sheahan et al., 2016*; *Schween et al., 2019*). In contrast, arbitrary visual features, such as color cues, have been generally ineffective (*Gandolfo et al., 1996*; *Karniel and Mussa-Ivaldi, 2002*; *Howard et al., 2012*; *Howard et al., 2013*; *Krouchev and Kalaska, 2003*; *Osu et al., 2004*; *Addou et al., 2011*; *Forano et al., 2021*). To account for this discrepancy, it has been proposed that movement-related cues are dynamically incorporated into the motor state, allowing for the separation of distinct internal models, whereas arbitrary cues are not sufficiently relevant to the motor state (*Howard et al., 2012*; *Howard et al., 2013*).

In the current study, we found compelling contextual effects using arbitrary sensory cues. Importantly, like delay eyeblink conditioning, these effects were subject to strong temporal constraint on the interval between the CS and US, similar to what is observed in other forms of cerebellar-dependent sensorimotor learning. Robust differential conditioning behavior was observed when the CS-US interval was short (Experiments 1 and 3); increasing the interval by just ~1000 ms abolished the Pavlovian effect (Experiment 2). These results are also consistent with prior evidence showing that the efficacy of movement-related cues is highly sensitive to timing (*Howard et al., 2012*).

These temporal constraints are likely built into standard adaptation tasks. The target is a salient stimulus that defines the task goal and movement plan, and its onset usually serves as the imperative for movement initiation. RTs in these tasks are typically below 500 ms (*Kim et al., 2018*; *Kim et al., 2019*; *Avraham et al., 2021*). As such, a tight temporal link is established between target appearance and movement, echoing (perhaps inadvertently) the CS-US temporal constraints essential for cerebellar-dependent conditioning. Under these conditions, the target can be viewed as a highly effective contextual cue.

More precisely, we propose that it is the movement plan itself that constitutes the primary, strongest CS. While the movement plan and target appearance usually coincide spatially (i.e. people aim to the target), this is not always the case. For example, with a standard visuomotor rotation, participants often deliberately aim away from the target, especially when the perturbation is large (*Hegele and Heuer, 2010*; *Taylor et al., 2014*). Recognizing this 're-aimed' plan as behaving like a CS may provide a parsimonious account of several phenomena. First, generalization of adaptation appears to be centered around the direction of the movement plan, not the target (*Day et al., 2016*; *McDougle et al., 2017*). Speculatively, this mirrors generalization effects seen in eyeblink conditioning, where variations on the learned CS (e.g. altering a tone's frequency) lead to parametric changes in CR probability (*Siegel et al., 1968*). These changes produce a similar Gaussian generalization function as that seen in adaptation generalization. Second, even in the absence of distinct contextual movements (e.g. follow-throughs), the mere activation of different motor plans can serve as efficient contextual cues, negating interference effects from opposing perturbations (*Sheahan et al., 2016*; *Sheahan et al., 2018*). Third, an emphasis on the movement plan is in accord with theoretical models of cerebellar-dependent adaptation, where the prediction that constitutes the basis for the sensory prediction error is computed using an efference copy of the movement intention (*Blakemore et al., 2001*; *Gao et al., 2016*; *Kawato and Gomi, 1992*; *Kim et al., 2022*; *Wolpert et al., 1998*).

We considered the possibility that the movement plan is a highly salient CS when designing the task. Participants were always moving to a single target location. Studies of motor adaptation commonly use multiple targets that span the entire workspace. However, the use of multiple targets could effectively increase the number of CSs in the task. By the additivity principle, these would compete with the arbitrary tone and light CSs, making it harder to observe their contribution to the Pavlovian effect. In addition, using a single target location allowed us to have the target remain on the screen throughout the entire experiment, eliminating any temporal linkage between the onset of the target and the feedback. While a single target may facilitate the use of simple explicit strategies (*McDougle and Taylor, 2019*), empirically, there is considerable evidence demonstrating that the clamp method effectively isolates implicit adaptation (*Morehead et al., 2017*; *Tsay et al., 2020c*; *Avraham et al., 2021*). Moreover, the Pavlovian effects were similar for the subset of participants who recognized the contingency between the CS and the US as for those who did not (see Results). Thus, while explicit strategies cannot be completely ruled out, we do not believe they significantly drove the observed effects.

Our focus in this study was on features of sensorimotor adaptation that map onto a CS, US, and CR. The UR in this context could be defined as an automatic action that is immediately triggered by the US, such as an online correction to the perceived error. While there is a rich literature showing the automatic and relatively fast latency of this corrective response (e.g. around 150 ms; *Carroll et al., 2019*; *Franklin and Wolpert, 2008*), to isolate feedforward adaptation, we opted to use rapid shooting movements that precluded online corrections (see Results). We recognize that a feedback response potentiated by the perturbation might be utilized for feedforward learning (*Albert and Shadmehr, 2016*). However, a UR is not always required for conditioning processes arising from volitional behaviors (e.g. *Cartoni et al., 2016*). Future work is needed to better understand the role of URs within our framework.

In terms of the feedforward response, there were notable differences in the extent of adaptation across the four experiments. Overall heading angle reached asymptotic values of ~15° in Experiments 1 and 4 and ~25° in Experiments 2 and 3. These differences are likely due to changes in the experimental platform. Experiments 1 and 4 were conducted in our laboratory, using a tablet and digitizing pen setup in which the movements involve arm movements of at least 8 cm. In contrast, Experiments 2 and 3 were conducted via a web-based platform, with each participant using their personal computer with a mouse or a trackpad to produce wrist or finger movements of ~2 cm (depending on the sensitivity of the device). Although basic phenomena of visuomotor adaptation are similar between lab- and web-based platforms (*Tsay et al., 2021*), there may be mechanistic differences. For example, with our laboratory setup, the monitor is positioned just above the hand. As such, the stimuli and movement are in parallel planes, and there is a 1:1 correspondence between the distance moved and the extent of the cursor movement. For the web-based platform, the stimuli and movement occur in roughly orthogonal planes, and there is a gain factor such that the cursor movement is larger than the hand movement. We expect that the laboratory setup may be much better suited for providing reliable information concerning proprioception, information that may limit the extent of adaptation (*Salomonczyk et al., 2011*; *Tsay et al., 2022*). Given that proprioceptive and visual signals are not aligned in the web-based studies, proprioception may have a reduced effect in limiting adaptation, leading to a higher asymptote.

To formally relate eyeblink conditioning and adaptation, we implemented the Rescorla-Wagner model, a classic associative learning model that has been widely employed in the classical and operant conditioning literature (*Rescorla and Wagner, 1972*). The success of this model to capture the general features of the current datasets should not be surprising since the model was developed to account for phenomena such as differential conditioning and compound conditioning. Thus, it was somewhat overdetermined that the Rescorla-Wagner model would provide better fits than the standard state-space model in our study, given that the latter cannot capture contextual effects. In theory, a standard state-space model could be modified such that different sensory cues become associated with different independent states to allow for context-dependent learning (*Heald et al., 2018*). However, we emphasize that to account for compound conditioning, and in particular the additivity principle (i.e. overshadowing), a state-space model would essentially have to be transformed into a facsimile of a Rescorla-Wagner model, supporting our key conclusions.

The Rescorla-Wagner model offers an account of how arbitrary sensory cues, visual targets, and movement plans can serve as CSs for adaptation. There are, however, some notable phenomena in the sensorimotor adaptation literature that are neither accounted for by our application of the Rescorla-Wagner model nor by the basic state-space model. Given the prominent role of the state-space model in the sensorimotor adaptation literature, variants of this model were developed to capture these observations. One such effect is spontaneous recovery, the re-manifestation of a previously adapted state in the absence of error feedback. A variant of the state-space model, one that allows for multiple states with different learning and forgetting rates, can capture spontaneous recovery (*Smith et al., 2006*). A second example is the relationship between learning rate and environmental consistency; learning is faster in response to a consistent vs inconsistent perturbation (*Albert et al., 2021*; *Avraham et al., 2020*; *Gonzalez Castro et al., 2014*; *Herzfeld et al., 2014*; *Hutter and Taylor, 2018*). To account for this effect, the state-space model can be modified to allow the learning rate to vary with experience (*Herzfeld et al., 2014*). A third example is savings upon relearning, where faster adaptation is observed upon a second exposure to the same perturbation (*Huang et al., 2011*; *Krakauer et al., 2005*). Studies have modeled savings using multi-rate state-space models (*Smith et al., 2006*), learning rate modulation (*Herzfeld et al., 2014*), or a combination of these factors (*Zarahn et al., 2008*; *Mawase et al., 2014*).

These higher-order phenomena have also been addressed in a new model of sensorimotor learning, one that takes a Bayesian inference approach (*Heald et al., 2021*). The contextual inference (COIN) model posits that motor adaptation arises from two interacting mechanisms: proper learning, which involves the creation and updating of context-specific memories, and apparent learning, a process of inferring the current context to determine which memory to express (recall). Unlike conventional context-independent models of motor adaptation, the COIN model stresses the importance of the environment in guiding memory retrieval. In one sense, our implementation of the Rescorla-Wagner model follows a similar philosophy. However,

it maps to the proper learning component of the COIN model since we assume that the current context is determined by the actual properties of the environment in an all-or-none manner (e.g. the CS presented by the experimenter). The COIN model suggests that spontaneous recovery, consistency effects, and savings may emerge due to contextual inferences; these inferences likely involve cognitive systems related to executive function and non-motor memory systems (*Collins and McDougle, 2021*). Consistent with this view, growing evidence shows that these phenomena are, in large degree, the result of strategic changes (*Haith et al., 2015*; *Morehead et al., 2015*; *Leow et al., 2020*; *Avraham et al., 2020*; *Avraham et al., 2021*; *Wang et al., 2022*).

While the basic Rescorla-Wagner model is unable to account for higher order effects such as spontaneous recovery or savings, we envision that complimenting it with more sophisticated, Bayesian inference models could readily accommodate these phenomena. In fact, models that combine associative mechanisms with some form of a Bayesian inference process that carves the world into distinct contexts have successfully captured a range of complex phenomena in classical conditioning and reinforcement learning (*Collins and Frank, 2013*; *Courville et al., 2006*; *Gershman, 2015*; *Kruschke, 2008*). We believe that our results lay the foundation for adopting a similar approach to study implicit sensorimotor adaptation and perhaps a more general account that captures the operation and interaction of multiple learning processes.

Motivated by the cerebellar literature, we limited our focus to delay eyeblink conditioning in considering how associative learning mechanisms can account for contextual effects in sensorimotor adaptation. We recognize that conditioning effects can be much more complex than the behavioral changes that arise from the pairing of two stimuli (*Rescorla, 1988*). These include associations between a CS and an action (*Skinner, 1938*), learning that does not abide by strict timing requirements (*Gallistel and Gibbon, 2000*), and other conditioning processes (e.g. trace conditioning) that rely on brain structures outside the cerebellum (*Clark et al., 2002*; *Clark and Squire, 1998*). Multiple learning processes allow the organism to build a comprehensive representation of the world, with the ability to flexibly modify behavior at many levels of control in response to changes in the contingencies between the body and environment (*McDougle et al., 2016*; *Kim et al., 2020*). Nonetheless, by showing that core constraints of delay eyeblink conditioning are relevant for sensorimotor adaptation, our study may inform a formal framework that looks at these two domains in a more unified manner.

Although our results highlight principles that address context-dependent sensorimotor adaptation, they do not speak to the error-correcting adaptation algorithm itself. Conventional models of this algorithm focus on the role of forward models in predicting future sensory states and updating motor commands to reduce sensory prediction errors (*Wolpert and Ghahramani, 2000*; but see *Hadjiosif et al., 2021*). Associative learning alone does not provide a mechanism for the error-correction process (e.g. the directional change of reaching movements given rotated feedback or the precise timing of an eyeblink response to a predicted air puff). Rather, it describes the association between a contextual cue and a salient event (e.g. shock, reward, etc.). Speculatively, an alternative idea is that sensorimotor adaptation may operate as a lookup table of context-outcome associations learned by a system designed to keep the sensorimotor system calibrated. A model-free conception like this could bring sensorimotor adaptation closer to classical associative models of cerebellar learning and plasticity (*Albus, 1971*; *Ito, 1984*; *Marr, 1969*).

## Materials and methods
### Participants
166 healthy volunteers (aged 18–35 years; 98 females, 6 identified as 'other') participated in either Experiment 1 (N=16), Experiment 2 (N=64), Experiment 3 (N=64), or Experiment 4 (N=22). In Experiments 1 and 4, all participants were right handed, as self-reported and verified with the Edinburgh Handedness Inventory. The participants in Experiments 2 and 3 did not complete a handedness inventory, but 108 of them self-reported being right handed, 17 left-handed, and 3 ambidextrous. The sample size for each experiment was set to ensure good statistical power (details provided in the protocol for each experiment). The protocol was approved by the Institutional Review Board at the University of California, Berkeley.

## Experimental setup and task

In Experiments 1 and 4, the participant sat at a custom-made table that housed a horizontally mounted LCD screen (53.2 cm by 30 cm, ASUS), positioned 27 cm above a digitizing tablet (49.3 cm by 32.7 cm, Intuos 4XL; Wacom, Vancouver, WA, USA). The participant held in their right hand a hockey paddle that contained an embedded digitizing stylus. The monitor occluded direct vision of the hand, and the room lights were extinguished to minimize peripheral vision of the arm. Reaching movements were performed by sliding the paddle across the tablet. The sampling rate of the tablet was 200 Hz.

For Experiments 2 and 3, we used a web-based platform that was created for conducting web-based studies in sensorimotor learning (*Tsay et al., 2020b*). Since participants performed the experiment remotely with their personal computer, the test apparatus varied across participants. Based on self-report data, 71 participants used an optical mouse, 55 used a trackpad, and 2 used trackball. Monitor sizes varied between 11 and 30 inches. The sizes of the stimuli were adjusted to the monitor size. Below we refer to the stimuli magnitudes in the lab-based experimental setup (Experiments 1 and 4).

At the beginning of each trial, a white circle (0.5 cm diameter) appeared at the center of the black screen, indicating the start location (*Figure 1A*). The participant moved the stylus to the start location. Feedback of hand position (i.e. the stylus position) was indicated by a white cursor (0.3 cm diameter), provided only when the hand was within 1 cm of the start location. A single blue target (0.6 cm diameter) was positioned 8 cm from the start location. In most studies of adaptation, the appearance of the target specifies both the movement goal (where to reach) and serves as the imperative (when to reach). From a classical conditioning perspective, the target should constitute a very salient CS given that its onset is temporally contingent with the US, the visual feedback associated with the movement (see below). To eliminate this temporal contingency, the target remained visible at the same location during the entire experiment. For each participant, the target was placed at one of four locations, either 45, 135, 225, and 315°, and this location was counterbalanced across participants.

The task included the presentation of neutral (non-spatial) CS(s). We used two different CSs, a tone and a light, both of which have no inherent association with the US. The tone CS was a pure sine wave tone with a frequency of 440 Hz. The light CS was a white rectangular frame (39.4 cm × 26.2 cm) that spanned the perimeter of the visual workspace. The large frame was selected to provide a salient visual stimulus but one that would not be confused with the target.

Depending on the specific experimental protocol and task phase, the CSs could appear alone or together on a trial. In Experiments 1 and 4, the onset of the CS served as the imperative signal, with the participant instructed to rapidly reach directly toward the target, slicing through the target. The onset of the CS occurred following a pseudo-random and predetermined delay after the hand was positioned at the start location. This was done to mitigate predictions regarding the timing of the CS onset. The delay ranged between 800 and 1200 ms (in steps of 100 ms), drawn from a uniform distribution. While the CSs were also presented in Experiments 2 and 3, a different signal was used for the movement imperative (see below). In all experiments, the CS was terminated when the hand reached the radial distance to the target (*Figure 1A*). In delay eyeblink conditioning, learning is optimal when the US is presented 100–500 ms after the CS (*Schneiderman and Gormezano, 1964*; *Smith et al., 1969*). To minimize the delay between the onset of the CS (or the other imperative cue) and the US, the auditory message 'start faster' was played whenever an RT exceeded 400 ms. RT was operationalized as the interval between the imperative onset and the time required for the radial distance of the hand to exceed 1 cm. Pilot work showed that RTs with the variable imperative onset time were always greater than 100 ms; as such, we did not set a minimum RT bound. Given our objective to test the link between feedforward adaptation and classical conditioning, we sought to eliminate online feedback corrections. Participants were instructed to make rapid movements, and the auditory message 'move faster' was played whenever MT exceeded 300 ms. The end of the movement was operationalized as the point where the radial distance of the hand reached 8 cm.

Experiments 2 and 3 were designed to test the hypothesis that the time interval between the arbitrary cues and the sensorimotor feedback is critical for driving Pavlovian effects. To this end, we did not use the CS onset as the imperative signal; rather, we introduced a new cue, a color change of the target from gray to blue, to serve as the imperative signal (*Figures 6A and 7A*). The difference between the two experiments was the interval between the onsets of the CS (tone or light) and imperative. In Experiment 2, the imperative was delayed with respect to CS onset, with the delay drawn

from a uniform distribution ranging from 800 to 1200 ms (in steps of 100 ms). As such, we added a mean of 1000 ms to the CS-US interval. The delay distribution that we used added variance to the CS-US interval, allowing us to make sure that, like Experiment 1, participants could not anticipate the onset of the imperative. This was important since predictability of the imperative timing could have decreased RTs and thus effectively decrease the delay between the CS and the feedback. We also note that previous evidence from eyeblink conditioning shows that the CR is minimally affected by the variability in the CS-US interval (*Patterson, 1970*). In Experiment 3, the CS and imperative were simultaneous (similar to the timing in Experiment 1). Trial durations were made comparable between Experiments 2 and 3 by setting the mean interval from the trial onset to the imperative in Experiment 3 to match that of Experiment 2.

For the US, we used task-irrelevant clamped feedback (*Morehead et al., 2017*). With clamped feedback, the radial position of the visual cursor is matched to the radial position of the hand. However, the angular position of the cursor is fixed. The participant thus controlled the speed and radial distance of the cursor, but not its direction. When designed to produce a prediction error and elicit implicit sensorimotor adaptation, the clamp followed a path that deviated from the target by 15°, with the direction, i.e., CW or CCW, counterbalanced across participants. We also included no-error trials (Experiments 1–3) by presenting clamped feedback that followed a path directly to the target (0° clamp; *Figure 2A*). The nature of the clamp manipulation was described in detail to the participant, and they were explicitly instructed strictly to ignore the feedback, aiming their reach directly toward the target on every trial. These instructions were designed to emphasize that the participant did not control the cursor's angular position and that they should always attempt to reach directly to the target. The instructions of the task were reinforced by the presentation of short video animations to demonstrate how the CSs would serve as imperative signals, as well as to show the invariant direction of the clamped feedback.

For the lab-based experiments (Experiments 1 and 4), the experimental software was custom written in MATLAB (The MathWorks, Natick, MA, USA), using the Psychtoolbox package (*Brainard, 1997*). The web-based experiments (Experiments 2 and 3) were developed for a webpage using JavaScript, HTML, and CSS. Data from the web-based experiments are temporarily stored via the Google Firebase database and downloaded for off-line analyses.

## Experimental protocol

All experiments included an acquisition phase and a probe phase. During the acquisition phase, clamped feedback was presented on each trial, serving as the US. During the probe phase, the clamped feedback was not presented. In both phases, the participants were instructed to reach straight to the target as soon as the imperative appeared. Note that we opted to not include baseline reaching blocks prior to the start of the acquisition phases to avoid introducing any incidental associations between the baseline feedback and the target, movement plan, and any other contextual variables. For the lab-based Experiments 1 and 4, a break of approximately 1 min was provided in the middle of the experiment.

## Experiments 1–3: differential conditioning

Experiment 1 (N=16) was designed to test differential conditioning in the context of a sensorimotor adaptation task. The target number of participants was determined based on the effect size (d=0.67) observed for the Pavlovian effect in a pilot study, with a significance level of $\alpha$=0.05 and power of 0.8. Experiments 2 and 3 were designed to test temporal constraints on differential conditioning. Participants (N=64 per experiment, 128 total) were recruited using the crowdsourcing website Prolific (www.prolific.co). The minimum sample size for each experiment was determined based on the sample size of Experiment 1 (N=16), where robust Pavlovian effects were observed. However, we have found that data collected with our web-based platform are more variable (*Tsay et al., 2021*). As such, we opted to use a larger sample size, setting our target as 64 participants for each experiment.

The session consisted of 800 trials: 600 acquisition trials followed by 200 probe trials (*Figure 2A*). One of two CSs (tone or light) was presented on each trial, serving as the imperative for the reaching response. During the acquisition phase, one CS was paired with 15° clamped error feedback (CS+ condition) and the other CS was paired with 0° clamped feedback (CS− condition). Each CS was

presented on 50% of the trials, and the assignment of the tone and light to the CS+ and CS− was counterbalanced across participants. During the probe phase, each CS was presented alone on half of the trials, and there was no visual feedback.

For both the acquisition and probe phases, the CS+ and CS− trials were interleaved in a pseudo-random order that was unique to each participant. To ensure that the participant would not be able to predict the CS type, the generated trial sequence for each participant was assessed to verify that there was no significant lag-1 autocorrelation in the time series. At the end of the experimental session, we assessed whether participants were aware of the contingency between each CS and its associated feedback, asking, 'did you identify any pattern in the experiment?' in a free-response survey.

## Experiment 4: compound conditioning

The adaptation task was modified in Experiment 4 (N=22) to provide a test of compound conditioning. The sample size was determined based on the effect size (d=0.56) observed for the Pavlovian effect in Experiment 1, with a significance level of $\alpha$=0.05 and power of 0.8.

The procedure was similar to that used in Experiments 1–3 with the following changes: first, the session consisted of 600 acquisition trials and 300 probe trials (*Figure 8A*); second, a compound CS, consisting of both the tone and light, served as the imperative throughout the acquisition phase, and was always paired with 15° clamped feedback; third, the probe phase (no visual feedback) consisted of 100 trials for each of the compound CS, tone alone CS, and light alone CS.

## Data analysis

The recorded position of the hand was analyzed using custom-written MATLAB scripts. Our main analyses focused on the reach direction (heading angle) and the trial-by-trial changes in heading angle (Δ heading angle). Heading angle was defined by two imaginary lines, one from the start position to the target and the other from the start position to the hand position at maximum movement velocity.

For all analyses, and to visualize the results, the sign of the heading angle was flipped for participants who experienced a CCW clamp, such that a positive heading angle is in the direction of expected adaptation (i.e. opposite the direction of the perturbed feedback). Moreover, the heading angle on the first acquisition trial was treated as the baseline reaching angle and subtracted from the heading angle on all subsequent trials (We note that the results remain unchanged in terms of statistical comparisons if this baseline subtraction step is omitted.).

For lab-based experiments (1 and 4), trials in which the heading angle was larger than 100° off from the target, or in which the trial-to-trial change in heading angle was larger than 25°, were considered outliers and not included in the analyses. For web-based experiments (2 and 3), outlier trials were considered trials in which the heading angle was beyond the range of −50–70° off the target, or in which the trial-to-trial change in heading angle was larger than 20°. The criteria for considering outlier trials were determined after visual examination of the time courses of all individuals in each experimental platform (lab-based and web-based). To eliminate biases, the CS was not visible in the trial traces. Outliers constituted 0.03 (Experiment 1), 1.26 (Experiment 2), 1.44 (Experiment 3), and 0.16% (Experiment 4) of the trials in each experiment.

For the change in heading angle analysis, but not for the presentation of heading angle time courses (*Figures 2B*, *6B*, *7B* and *8B*), we excluded trials in which the RT exceeded 400 ms and/or MT exceeded 300 ms (Experiment 1: 11% of acquisition trials and 4.3% of probe trials; Experiment 2: 19% of acquisition trials and 16% of probe trials; Experiment 3: 19% of acquisition trials and 15% of probe trials; Experiment 4: 3.4% of acquisition trials and 4.9% of probe trials).

In Experiments 1–3, the primary analyses examined how the Δ heading angle was influenced by the CS type (CS+ vs CS−), either in terms of the previous trial (*n*−1, adaptation effect) or current trial (*n*, Pavlovian effect). For each participant and phase (acquisition or probe), we calculated the average Δ heading angle for four types of trials: CS+ trials that follow CS+ trials, CS− after CS+, CS+ after CS−, and CS− after CS−. For each phase, a two-way repeated-measures ANOVA was conducted with two within-participant independent factors, the CS on trial *n*−1 and the CS on trial *n*, each with two levels, CS+ and CS−. The Δ heading angle was the dependent variable in the two ANOVAs. To examine the dynamics of

the adaptation and Pavlovian effects (*Figure 5B*), we binned the heading angle data into epochs of 50 trials. Within each bin, we performed a multiple regression analysis to test whether trial-to-trial changes in heading angle can be predicted from the previous trial CS, the current trial CS, and their interaction. *Figure 5B* presents the mean ± SEM regression $\beta$ weights of all simulated time courses for the previous (adaptation effect) and current (Pavlovian effect) predictors. To evaluate statistically the changes in $\beta$ weights for each type of effect across bins, we used a linear mixed model (R statistical package: lmerTest), with type (adaptation and Pavlovian) and bin as fixed effects and participants as random effects.

To assess whether the Pavlovian effect in Experiment 1 is influenced by the awareness about the contingency between each CS and its respective feedback, we divided the participants based on their responses to the post experiment questionnaire (see Experimental protocol): 7 out of the 16 participants stated the correct contingency between the CS and the visual feedback and were thus considered the 'aware sub-group.' The rest of the participants (N=9) reported that they did not identify any pattern related to the CS-US contingency and were considered the 'unaware sub-group.' Independent two-sample *t* tests were used to compare the Pavlovian effects between these groups during the acquisition and probe phases of the experiment.

Feedback correction was operationalized as the difference between the heading angle measured at the radial distance to the target and at 50 ms after movement initiation. We estimated the mean and SD of feedback correction across all acquisition trials in all the participants in Experiment 1. In addition, we calculated, for each participant, the mean feedback correction for each of the CS+ and CS− trials and used a paired-sample *t* test to examine within-participant changes in feedback correction between the two trial types.

In Experiment 4, the analysis focused on the probe phase in which there was no visual feedback. We compared the Δ heading angle in response to the three CSs on trial *n* (compound CS, light alone, tone alone) regardless of the CS presented on trial *n*−1. As the assumption of normality was not met for the distribution of one of the conditions, we used the non-parametric Friedman test, with the Δ heading angle as the dependent variable, and the CS type as the within-participant independent variable.

The additivity principle of the Rescorla-Wagner model states that the elements of the compound CS will compete with one another for the strength of association with the US (*Equation 3*). This principle was tested in our data by examining the correlation between the Δ heading angle associated with the tone and light CSs in Experiment 4.

We report effect size using Cohen's *d* for all *t* tests, partial eta-squared ($\eta_p^2$) for the ANOVA and the Kendall's W for the Friedman test. We followed the assumption of normality when the sample size was larger than N=30 based on the central limit theorem or when the Lilliefors test indicated normality (*Lilliefors, 1967*). When these conditions were met, we used parametric statistical tests. In all other cases, we used non-parametric statistics (see Experiment 4). Family-wise errors in pairwise comparisons were corrected using the Bonferroni correction. We also report Bayes factor $BF_{10}$, the ratio of the likelihood of the alternative hypothesis ($H_1$) over the null hypothesis ($H_0$) (*Kass and Raftery, 1995*).

## Model simulations

Trial-by-trial reach angles were simulated using the Rescorla-Wagner model as a representative model for associative learning. In this model, the motor state is updated based on the associative strength (*V*) between the US (error signal) and all the CSs present on a given trial (*Equations 1* and *3*). The extent of learning is determined by the maximum conditioning level ($\lambda$), and the rate of behavioral change is determined by the learning rate of the US ($\beta$) and the salience of each presented CS (*a*). Note that in simulating the Rescorla-Wagner model for visuomotor adaptation, we included the movement plan as a CS (CS_plan) on all trials in addition to the trial-specific CS (CS_tone or CS_light). The following pseudocode describes how the heading angle, represented by the total associative strength V_total between all CSs and the US, is updated in a simulation of differential conditioning. For simplicity, we consider CS_tone = CS+ and CS_light = CS−. US = 1 on CS+ trial and US = 0 on CS− trials and on no feedback trials (all the trials in the probe phase).

On trial n=1, before experiencing the US, all associative strengths are initialized to 0.
FOR each other trial n in the experiment
 $V\_plan_n \leftarrow V\_plan_{n-1} + \alpha\_plan \times \beta \times (\lambda \times US_{n-1} - V\_total_{n-1})$
 IF trial n−1 is CS_tone
 $V\_total_{n-1} \leftarrow V\_plan_{n-1} + V\_tone_{n-1}$
 $V\_tone_n \leftarrow V\_tone_{n-1} + \alpha\_tone \times \beta \times (\lambda \times US_{n-1} - V\_total_{n-1})$
 $V\_light_n \leftarrow V\_light_{n-1}$
 ELSE IF trial n−1 is CS_light
 $V\_total_{n-1} \leftarrow V\_plan_{n-1} + V\_light_{n-1}$
 $V\_light_n \leftarrow V\_light_{n-1} + \alpha\_light \times \beta \times (\lambda \times US_{n-1} - V\_total_{n-1})$
 $V\_tone_n \leftarrow V\_tone_{n-1}$
 END IF
END FOR

In compound conditioning, all the trials in the acquisition phase are presented with an error (US = 1) and with both CS_tone and CS_light (and CS_plan). Therefore, both V_tone and V_light (and V_plan) are updated on all trials:

FOR each trial n>1 in the acquisition phase
 $V\_total_{n-1} \leftarrow V\_plan_{n-1} + V\_tone_{n-1} + V\_light_{n-1}$
 $V\_plan_n \leftarrow V\_plan_{n-1} + \alpha\_plan \times \beta \times (\lambda \times US_{n-1} - V\_total_{n-1})$
 $V\_tone_n \leftarrow V\_tone_{n-1} + \alpha\_tone \times \beta \times (\lambda \times US_{n-1} - V\_total_{n-1})$
 $V\_light_n \leftarrow V\_light_{n-1} + \alpha\_light \times \beta \times (\lambda \times US_{n-1} - V\_total_{n-1})$
END FOR

During the probe phase in compound conditioning, the associative strengths are updated based on the CS presented on each trial, either as a compound, which is simulated the same as the above pseudocode for compound conditioning, or alone, and then the algorithm uses the same implementation as in differential conditioning. We set US=0 in all the trials of the probe phase.

To illustrate the predictions of the model in *Figures 2E, F and 8D*, we chose parameters that result in qualitatively similar effects to the experimental results. For both Experiments 1 and 2, we set $\lambda$ to 15, similar to the observed implicit adaptation asymptotes in these experiments, as well as in other studies (*Bond and Taylor, 2015*; *Morehead et al., 2017*). The value of $\beta$ was set to 0.12 (Experiment 1) or 0.02 (Experiment 4), the salience parameter ($\alpha$) for the movement plan CS to 0.99, and both the tone CS and light CS to 0.002 (Experiment 1) or 0.1 (Experiment 4). These divergent salience values are consistent with the assumption that most of the associative strength of the US would be absorbed by the movement plan CS given its central relevance to the task of reaching to a target. The remainder of potential associative strength to the US is thus available for the tone and light CSs.

To demonstrate how the choice of parameters in the Rescorla-Wagner model influences the predicted Pavlovian effects, we simulated behavior using various combinations of the free parameters. We chose several combinations of values for $\beta$ and $\lambda$, and, for each combination, a wide range of values for $\alpha_{goal}$, $\alpha_{tone}$ and $\alpha_{light}$ (For simplicity, we constrained $\alpha_{tone}$ and $\alpha_{light}$ to the same value.). *Figure 3* displays heatmaps resulting from these simulations, with the color of each cell corresponding to the simulated difference in the trial-by-trial change in heading angle between CS+ and CS− trials. In addition, we illustrated that a single-process state-space model, a standard model of sensorimotor adaptation, cannot capture these Pavlovian effects. In this model, the motor state is updated according to the error observed during the previous movement. The parameters of this model determine both the trial-by-trial retention of the previous motor state and the rate of learning from errors (see *Equation 2* in the Results). For the simulation presented in *Figure 3*, the error was fixed to one of two values during the acquisition phase, either −15° on CS+ trials or 0° on CS− trials, and in the probe phase, it was set to 0° on all trials.

While the two models share similar features and parameters (e.g. learning rate of an error signal), the additional parameters in the Rescorla-Wagner model allow it to capture the effects of differential and compound conditioning, should these processes be operative in our experiments. For example, in the differential conditioning case, modifying the parameters of the Rescorla-Wagner model could produce changes in the magnitude of both the within-trial 'Pavlovian' effects (i.e. the modulation of behavior in response to the current CS, or 'trial *n*' effects) and across-trial adaptation effects (i.e. the modulation of behavior in response to the current state of learning, or 'trial *n*−1' effects). However, the Rescorla-Wagner model will always yield unique responses to the CS+ and CS− given non-zero

salience parameters. In contrast, no combination of the *A* and *B* parameters in the state-space model will produce differential responses to the tone and light CSs.

To simulate the dynamics of the adaptation and Pavlovian effects in differential conditioning (*Figure 5A*) and to compare it to the dynamics observed in the actual data (*Figure 5B*), we simulated the trial-by-trial change in heading angle during acquisition (600 trials) according to the Rescorla-Wagner model (*Equations 1; 3*) and the state-space model (*Equation 2.*) based on the schedules of CS+ and CS− trials that were presented to the participants (16 simulated time courses). For the Rescorla-Wagner model, we used the same parameters values that generated the simulation results presented in *Figure 2E and F*. For the state-space model, we set the *A* parameter to 0.9 and *B* to 0.12. We then did the same trial-by-trial regression analysis described above for the actual heading angle data (see Data analysis) with the exception that the dependent variable was now the simulated change in heading angle.

## Model fitting and comparison

We conducted a post-hoc model comparison analysis (*Figure 4*). In this analysis, we fit the Rescorla-Wagner model (*Equations 1; 3*) and the standard state-space model (*Equation 2*) to participants' heading angle time course data of Experiment 1. The two models were fit by minimizing the sum of squared residuals between the measured and modeled movement data, using the MATLAB function fmincon. To avoid local minima, 200 randomized sets of initial parameter values were used during fitting, and the best fit of each model was selected for model comparison. Models were compared using both the sum of squared residuals and the AIC approximated on the residuals (*Akaike, 1974*). All free parameters were bound at (0, 1), except for $\lambda$, which was bounded at (–30, 60). We did not perform model fitting for Experiment 4 since all trials were identical in the acquisition phase. As such, the salience parameters in the Rescorla-Wagner model for the goal, tone, and light are unidentifiable, and both models make indistinguishable behavioral predictions during acquisition.

## Acknowledgements

We thank members of the BLAM lab (Johns Hopkins) for helpful discussions. We thank Marina Iranmanesh, Janet Hwang, Sarvenaz Pakzad and Aaron Shalf for their assistance with data collection. JAT is funded by the NIH (NS084948), the NSF (1838462 and 1827550), ONR N00014-18-1-2873, and J Insley Blair Pyne Fund. RBI is funded by the NIH (NS116883 and DC077091).

## Additional information

### Competing interests

Richard B Ivry: Senior editor, *eLife*. The other authors declare that no competing interests exist.

### Funding

| Funder | Grant reference number | Author |
| --- | --- | --- |
| National Institutes of Health | NS084948 | Jordan A Taylor |
| National Science Foundation | 1838462 | Jordan A Taylor |
| National Science Foundation | 1827550 | Jordan A Taylor |
| Office of Naval Research | N00014-18-1-2873 | Jordan A Taylor |
| National Institutes of Health | NS116883 | Richard B Ivry |
| National Institutes of Health | DC077091 | Richard B Ivry |

| Funder | Grant reference number | Author |
|---|---|---|

The funders had no role in study design, data collection and interpretation, or the decision to submit the work for publication.

## Author contributions
Guy Avraham, Conceptualization, Data curation, Software, Formal analysis, Validation, Investigation, Visualization, Methodology, Writing – original draft, Project administration, Writing – review and editing; Jordan A Taylor, Conceptualization, Supervision, Methodology, Writing – review and editing; Assaf Breska, Conceptualization, Writing – review and editing; Richard B Ivry, Conceptualization, Resources, Supervision, Funding acquisition, Project administration, Writing – review and editing; Samuel D McDougle, Conceptualization, Software, Supervision, Methodology, Writing – original draft, Project administration, Writing – review and editing

## Author ORCIDs
Guy Avraham http://orcid.org/0000-0002-6170-1041
Jordan A Taylor http://orcid.org/0000-0001-9300-1229
Assaf Breska http://orcid.org/0000-0002-6233-073X
Richard B Ivry http://orcid.org/0000-0003-4728-5130
Samuel D McDougle http://orcid.org/0000-0001-8100-4238

## Ethics
Human subjects: The study was approved by the Institutional Review Board at the University of California, Berkeley (Protocol 2016-02-8439) and adhered to the principles expressed in the Declaration of Helsinki. All participants provided written informed consent to participate in the study.

## Decision letter and Author response
Decision letter https://doi.org/10.7554/eLife.75801.sa1
Author response https://doi.org/10.7554/eLife.75801.sa2

# Additional files

## Supplementary files
• Transparent reporting form

## Data availability
All data generated or analysed during this study are included in the manuscript; Source Data files have been provided for Figures 2B, 2C, 2D, 5B, 6B, 6C, 6D, 7B, 7C, 7D, 8B, 8C, 8E and 8F. All raw data files and codes for data analysis and simulations are available from the GitHub repository: https://github.com/guyavr/AssociativeMotorAdaptation.git (copy archived at swh:1:rev:f0c5c7cb930dc7952c24021113edac8be7e4bf32).

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
