## [Editor Report]

This paper provides a fundamental account of the role associative learning plays in sensorimotor adaptation. In a compelling result, the authors show that by pairing movement-related feedback with conditioning cues in the form of neutral auditory or visual contextual cues can be used to differentiate between sensorimotor perturbations/states. This work nicely integrates multiple literatures surrounding the processes supported by the cerebellum and solves a long-standing puzzle of exactly how and when arbitrary cues can serve to shape motor adaptation.

---

## [Decision Letter]

**Decision letter after peer review:**

Thank you for submitting your article "Contextual effects in sensorimotor adaptation adhere to associative learning rules" for consideration by *eLife*. Your article has been reviewed by 2 peer reviewers, and the evaluation has been overseen by a Reviewing Editor and Michael Frank as the Senior Editor. The reviewers have opted to remain anonymous.

Essential revisions:

1) More evidence for the temporal specificity of results. The authors advance the idea that the key difference between the current results and previous work that has failed to find evidence of associative learning is the temporal relationship between cues and feedback. However, no experimental evidence is provided that this timing is fundamental. There is a need for new data showing the importance of the temporal relationship to support many of the conclusions drawn in the article. (See comments from reviewer 2 below.)

2) There are many instances of places in the manuscript that lacked clarity and could benefit from more exposition and inclusion of some of the supplementary analyses. These include a better explanation of the task, how the various stimuli (US, CS, etc) map onto the task, the inclusion of Supp. Figure 1 in the main text, and (See comments from both reviewers below.)

3) More discussion about the models considered and their similarities/differences and their diverging predictions.

*Reviewer #1 (Recommendations for the authors):*

I recommend writing a clearer explanation of the associative learning approach investigated here and the relationship between standard ways of looking at visuomotor rotation and associative learning and making the notation for the latter clear and concise

I suspect some of the readers of this article will be more familiar with VR rotation and dynamic learning as less so with classical conditioning. The task could certainly be explained more clearly.

It would help to have a more explanation of how the current study operates as a standard visuo motor rotation task. Then explain how this is related to a classical conditioning task. There are lots of abbreviations including CS+, CS- etc. it would be helpful to have a table to illustrate the analogous link between conditioning and VR learning.

Figure 1 could be explained better. Suggest having a bigger schematic of the task since understanding this is key to understanding the article. maybe have it as a single figure

Simulations of the RW models are carried out. However only their basic formulas are stated (Equations 1 and 2). It would be better to show more details in the methods (by providing the equations) for how precisely experiments 1 and 2 were simulated. pseudocode for explaining simulations would also be nice.

There is a mention of simple state space models not being able to deal with contextual effects. It would be useful to elaborate on this with reference to the equations that describe these models. It would also be helpful to discuss the differences between RW model and state space models, And also the deficiencies of both.

Supplementary figures are used. I would suggest it's better to fully explain them and include them as part of the main document.

The Github supplementary materials appears to be a clone of the author's working directory for simulations of this project and it is also hard to follow. There is of course no need for all this detail, but it's fine to provide it. I would however suggest that the main scripts for critical simulations for Experiments 1 and 2 are made easy to identify and run, if readers wish to do so.

*Reviewer #2 (Recommendations for the authors):*

Temporal specificity of results: The authors predict and conclude that the key parameter for observing associative learning when using arbitrary stimuli as contextual cues is the temporal relationship of these cues to the feedback. They provide compelling evidence of associative learning however at present provide no evidence of the importance of this temporal relationship. Considering this is assumed to be the key difference between the current results and previous work who have failed to find an effect, I believe the article requires additional work (maybe by introducing a delay between CS and movement) that examines the importance of this temporal relationship. Providing evidence that the associative learning effect is dependent on the timing between the cue and feedback being between 100-500ms is fundamental to support some conclusions made in the article.

Exp 1 probe phase results: Whilst not significant (n-1 x n interaction), there seemed to be a clear difference between CS+ trials within the different trial n-1 contexts (Figure 1D). In fact, this difference seems bigger (and as consistent) as the meaningful/significant differences which are focused on. Interestingly, the RW model predicts a clear n-1 x n interaction however it is not discussed why (Figure 1F). To me it seems that the behaviour (at least partially) and model reflect an interaction between trial n-1 and n during probe trials however this is currently not discussed. Could the authors elaborate on this result and include this in the article?

Magnitude of effect (supplementary S1 figure): Suppl Figure 1 needs to be in the main article as it provides context and some 'raw' data. It would also be beneficial to have in the main article a similar figure but with the CS+ and CS- trial types separated. Suppl Figure 1 is important as it highlights that the conditioning effects are relatively small. This needs to be explained/mentioned in the results i.e., that the differences between CS+ vs CS- (1-degree) are approx. 6% of the total adaptation that occurred (15-degrees). I suppose this is referred to within the final exp 1 analysis (Figure 2B) but its not explicit. Although I believe the results are important, the article currently reads as if these conditioning effects were large when in fact other people might conclude that conditioning had little impact on adaptation (as similar adaptation (approx. 15-degrees) was observed across both contexts (assumed as this is not shown) and performance looks very similar to exp 2 where there was no 0-degree context). There needs to be some acknowledgement of the fact that while these conditioning effects appear meaningful, they were small (with most participants showing less than a 1-degree difference between contexts).

Exp 2 results: It was unclear to me why the RW model would predict a negative heading angle within the single CS conditions (Figure 3C)? I understand that a weaker conditioning response would be expected due to compound conditioning, but would you not expect this still to be positive? Why would the model predict extinction to occur and why is this seen in the behaviour? The details of this result (and the predictions of the model) are currently not discussed in sufficient/any detail.

What is the CS (confusion between results and discussion)? Between lines 395-410 the authors describe the primary CS as being the heading angle ('the movement plan itself, rather than the target cue, that constitutes the primary CS'), however in the results (lines 74-96) they describe the CS as being the arbitrary cue ('When considered through the lens of classical conditioning, the arbitrary cues are the conditioned stimuli (CSs)') and the CR being the heading angle ('the conditioned response (CR) would be the movement heading angle elicited by a CS'). As a result of this discrepancy, I found this section of the discussion very confusing (lines 387-410 and then again from line 433). Are the authors saying that sometimes the heading angle/plan is the CS and other times it is the CR…? How does this all align? One can see why a suggestion for a figure showing this mechanism visually is suggested below.

Link between this work and recent contextual inference model by Heald et al.,: Could the authors provide a more direct comparison between the current work and the contextual inference model by Heald et al., (in discussion)? The authors currently say 'their model suggests that spontaneous recovery and consistency effects emerge due to contextual inferences that likely interact with deliberate changes in explicit strategies' however does this align with the current work? How does contextual inference align/differ with these conditioning mechanisms? In future work, the authors seem to want to explain the phenomena (spontaneous recovery etc) recently explained by this model and I am interested to know whether these are competing explanations or are explaining the same mechanism or are different but complementary?

Use of supplementary figures: Why did the authors decide to put so much of the important detail into the supplementary? In my opinion, all of this should be in the main article.

Normality of data and presentation of individual data: There is no mention of any assessment of data normality, was the data normally distributed? In addition, a greater amount of individual data should be shown rather than mean +- SEM.

Figure to represent conditioning mechanism: I would find it very helpful if an additional figure was added which showed the US, UR, CS and CR visually. I kept forgetting how each of them were proposed to be represented in the task.

---

## [Author Response]

Essential revisions:1) More evidence for the temporal specificity of results. The authors advance the idea that the key difference between the current results and previous work that has failed to find evidence of associative learning is the temporal relationship between cues and feedback. However, no experimental evidence is provided that this timing is fundamental. There is a need for new data showing that examines the importance of the temporal relationship to support many of the conclusions drawn in the article. (See comments from reviewer 2 below.)

We thank the reviewers for raising this important concern. We agree that temporal specificity is an important issue given the parallelism we wish to make between adaptation and eyeblink conditioning. We have collected new data to examine this question.

We designed a new variant of Experiment 1, our differential conditioning experiment. To highlight the change, we provide a schematic comparing the events in Experiment 1 and the new Experiment 2 (Author response image 1). (The schematic for the new Exp 2 appears in the new Figure 6A in the revision). In Experiment 1, the tone and light CSs served as the imperative signal (Author response image 1). By requiring a rapid response to the CS, we created a tight temporal link between the cue (CS) and the movement (and thus also the feedback, the US). In the new experiment (reported as Experiment 2 in the revision), we modified the task such that the movement imperative was signaled by a separate cue, a color change of the target from gray to blue (Author response image 1). This subtle alteration allowed us to impose a delay between the CS onset and imperative cue. The delay was pseudo-randomly drawn from a uniform distribution of values between 800-1,200 ms (in steps of 100 ms). As such, the timing of the imperative was unpredictable (like in Experiment 1), and critically, the delay values were beyond what is typically considered optimal for cerebellar-dependent conditioning. We hypothesized that, like prior studies (e.g., Howard et al. 2012, 2013), the Pavlovian effect (or trial *n* effect) observed with a shorter CS-US interval (i.e., the ~300 ms reaction time; Experiment 1, Figure 2) would now be diminished or undetectable.

**Author response image 1. sa2fig1:** Experiment 2: task design (A) The task design of Experiment 1 where the CS (tone or light) was the imperative signal. (B) Task design of Experiment 2. To test the temporal specificity of the Pavlovian effect, a color change of the target served as the imperative. This allowed us to impose a delay of ~1,000 between the CS and US.

Due to a surge in COVID cases around the time of our revisions, we ran the experiment online, using a web-based platform we have developed for sensorimotor learning studies (OnPoint, Tsay et al. 2021, *PsyArXiv*). The in-person sample size in Experiment 1 was N=16. For the new experiment, we collected data from 64 participants to maximize power. Our experience is that the data from web-based experiments is more variable than in in-person experiments. This may be due to variation in equipment (for web-based experiments, participants use their own laptops), movement effector (wrist or finger movements using a mouse or a trackpad, respectively), and supervision (instructions are provided through on screen messages) (Tsay et al. 2021, *NBDT*).

The results supported our hypothesis: We did not detect a significant Pavlovian effect in either the acquisition phase or the probe phase. (The results for the new Exp 2 appears in Figure 6 in the revision).

Please note that during the acquisition phase, one participant exhibited a negative Pavlovian effect with a value far smaller than the other participants (thus the broken y-axis in the right panel of Figure 6C). We opted to include this participant for two reasons. First, the participant’s data was not excluded by our a priori criteria for removing outliers. Second, the participant’s Pavlovian effect during the probe phase fell within the distribution of scores. Nonetheless, we did conduct a post-hoc analysis in which we excluded this participant. In this analysis, the Pavlovian effect was significant during the acquisition phase ([F(1,62)=4.57, p=0.037, BF_10_=1.18, η_p_^2^=0.06]) but not during the probe phase test ([F(1,62)=2.39, p=0.127, BF_10_=0.42, η_p_^2^=0.04]). As noted in the manuscript, the probe phase provides the cleanest assessment of the Pavlovian effect. This is underscored by the Bayes Factor (BF_10_) values. While this value is 2.89 in the acquisition phase of Experiment 1, it rises to 72.15 in the probe phase (the same pattern is seen in Experiment 3, described below).

In summary, the key takeaway we draw from this new experiment is that increasing the CS-US interval diminishes the efficacy of arbitrary cues in context-dependent motor adaptation. This pattern echoes what has been observed with cerebellar-dependent eyeblink conditioning (Schneiderman and Gormezano 1964, Smith et al. 1969).

We recognize that in this new experiment, participants no longer needed to attend to the tone and light CSs since these no longer served as the movement imperatives; as such, this may have reduced the salience of these cues. To address this concern, we conducted another new experiment, reported as Experiment 3 in the revised manuscript. The experiment was identical to Experiment 2 (including being web-based) with one key change: The imperative and CS were presented simultaneously (Figure 7A in the revision). Thus, while the new imperative cue was still the explicit “go” signal, there was no added delay between the CS and the imperative. We predicted that this should result in a resurrection of the Pavlovian effects. To match the trial durations of Experiments 2 and 3, we increased the interval between the onset of the trial and onset of the imperative/CS.

When we eliminated the extended delay between the CS and the US, we again observed robust differential conditioning (Pavlovian) effects, replicating the main pattern of behavior observed in Experiment 1 (Figures 7C, 7D in the revision). Thus, the results of Experiment 3 provide an additional demonstration that arbitrary cues can be associated with distinct sensorimotor outcomes. When considered in light of the results of Experiment 2, we see that this associative capability is subject to temporal constraints. The results of Experiment 3 are especially compelling to us given that participants were not explicitly directed to attend to the tone and light CSs since they no longer served as the imperative.

The results of both new experiments are now included in the revised manuscript (Figures 6 and 7). In the original manuscript, we had assumed that a relatively short interval between the CS and US was required for the Pavlovian effect. These new data put that assumption to test, with the results providing support for that assumption and for our main conclusions concerning the re-framing of sensorimotor adaptation as a form of conditioning. We think that these additional experiments greatly strengthen the paper, and we thank the reviewers for motivating this further work.

2) There are many instances of places in the manuscript that lacked clarity and could benefit from more exposition and inclusion of some of the supplementary analyses. These include a better explanation of the task, how the various stimuli (US, CS, etc) map onto the task, the inclusion of Supp. Figure 1 in the main text, and (See comments from both reviewers below.)

We have made changes to the manuscript with an eye to improving clarity. These include:

– Revising the description of the task and clarifying how it could map onto the components of classical conditioning.

– Inclusion of new Figure 1.

– We now include all supplementary figures in the main text.

Further details of the revisions are provided in the point-by-point response to the specific comments of the reviewers.

3) More discussion about the models considered and their similarities/differences and their diverging predictions.

We revised the paper to include a more direct comparison between the Rescorla-Wagner model and the state space model. We now provide detailed pseudocode to clarify the implementation of the Rescorla-Wagner model and discuss its similarities and differences with respect to other models in the field (multi-rate state space models, memory of error, and context inference).

Further details are provided below.

Reviewer #1 (Recommendations for the authors):I recommend writing a clearer explanation of the associative learning approach investigated here and the relationship between standard ways of looking at visuomotor rotation and associative learning and making the notation for the latter clear and conciseI suspect some of the readers of this article will be more familiar with VR rotation and dynamic learning as less so with classical conditioning. The task could certainly be explained more clearly.It would help to have a more explanation of how the current study operates as a standard visuo motor rotation task. Then explain how this related to a classical conditioning task. There are lots of abbreviations including CS+, CS- etc. it would be helpful to have a table to illustrate the analogous link between conditioning and VR learning.Figure 1 could be explained better. Suggest having a bigger schematic of the task since understanding this is key to understanding the article. maybe have it as a single figure

We thank the reviewer for the constructive feedback. We have revised our explanation of the task. This includes changes in the Introduction:

“In this form of classical conditioning [delay eyeblink conditioning], a sensory cue (conditioned stimulus, CS, e.g., a tone) is repeatedly paired with an aversive event (unconditioned stimulus, US, e.g., an air puff to the cornea). By itself, the aversive event naturally causes an immediate unconditioned response (UR, e.g., an eye blink in response to the air puff). After just a short training period with a reliable CS, organisms as diverse as turtles and humans produce an adaptive conditioned response (CR, e.g., an eye blink that anticipates the aversive air puff).”

and Results sections:

“In typical adaptation tasks, the onset of the target serves as the imperative for movement initiation. When considered through the lens of classical conditioning, the target appearance could be viewed as a conditioned stimulus (CS) given that it is presented just before the sensory feedback (the unconditioned stimulus, US) resulting from the movement (Figure 1B). To test the efficacy of arbitrary cues as conditioned stimuli, the onset of the tone or light served as the imperative, with the target visible at its fixed location throughout the experimental session (Figure 1B)…”

“… To continue with our conditioning analogy, the conditioned response (CR) would be the movement heading angle elicited by a CS, expressed as a change in heading angle following the pairing of that CS with the visual error, the US (Figure 1B). We note here that the unconditioned response (UR) in an adaptation task could correspond to an immediate movement correction in response to the perturbation; that is, an “online” correction. Given that we required fast movements to minimize these corrections, this potential form of UR is not observed in our design.”

In addition, we now have a figure devoted to providing a schematic of the task which includes how the task components could be described in a classical conditioning framework (Figure 1). We have also added Figures 6A and 7A that describe the task schematics of our new experiments (Experiments 2 and 3), demonstrating how the timing of the task events is critical for observing differential conditioning during sensorimotor adaptation.

Simulations of the RW models are carried out. However only their basic formulas are stated (Equations 1 and 2). It would be better to show more details in the methods (by providing the equations) for how precisely experiments 1 and 2 were simulated. pseudocode for explaining simulations would also be nice.

We added more details on how the simulations of the RW model were carried out and now provide the pseudocode (*Materials and methods*).

There is a mention of simple state space models not being able to deal with contextual effects. It would be useful to elaborate on this with reference to the equations that describe these models. It would also be helpful to discuss the differences between RW model and state space models, And also the deficiencies of both.

We have moved the description of the state space model from the *Materials and methods* to the *Results* section such that it now appears shortly after the description of the Rescorla-Wagner model. We discuss similarities and differences between the models and detail their shortcomings in the *Results* , *Discussion* and *Materials and methods* sections.

Supplementary figures are used. I would suggest it's better to fully explain them and include them as part of the main document.

We now include all supplementary figures as part of the main text.

The Github supplementary materials appears to be a clone of the author's working directory for simulations of this project and it is also hard to follow. There is of course no need for all this detail, but it's fine to provide it. I would however suggest that the main scripts for critical simulations for Experiments 1 and 2 are made easy to identify and run, if readers wish to do so.

We thank the reviewer for this suggestion. We have reorganized the Github repository:

https://github.com/guyavr/AssociativeMotorAdaptation.git

All data files, data analysis and simulation codes (including those for the two new experiments), are now arranged in separate folders, and we provide a detailed ReadMe file to make navigation easier.

Reviewer #2 (Recommendations for the authors):Temporal specificity of results: The authors predict and conclude that the key parameter for observing associative learning when using arbitrary stimuli as contextual cues is the temporal relationship of these cues to the feedback. They provide compelling evidence of associative learning however at present provide no evidence of the importance of this temporal relationship. Considering this is assumed to be the key difference between the current results and previous work who have failed to find an effect, I believe the article requires additional work (maybe by introducing a delay between CS and movement) that examines the importance of this temporal relationship. Providing evidence that the associative learning effect is dependent on the timing between the cue and feedback being between 100-500ms is fundamental to support some conclusions made in the article.

The reviewer is correct that we assume the associative phenomena we report are subject to strong temporal constraints. Indeed, this assumption is core to our claim about why prior studies generally failed to find adaptation in response to arbitrary sensory cues. As exemplified by eyeblink conditioning, the strength of the association (at least for many skeletal responses) is highly dependent on the CS-US interval (Schneiderman and Gormezano 1964, Smith et al., 1969). In reviewing previous studies that did not find contextual effects of arbitrary cues on motor adaptation, we saw that the interval between cue onset and movement initiation was around 1,000 ms in some studies, or was uncontrolled in others (Howard et al., 2012, Howard et al., 2013). These conditions are suboptimal for cerebellar-dependent conditioning.

In the experiments presented in the initial submission, we imposed temporal control by having the CSs act as movement imperatives and enforcing rapid (<400ms) reaction times. In this way, we reduced the CS-US interval to around 300 ms.

As noted by the reviewer, we did not include a negative control, a condition to show that the Pavlovian effects are absent (or attenuated) when these temporal requirements are not met. To address this concern, we have conducted two new experiments. The details are provided above (Essential Revisions). In brief, we no longer had the CS onset serve as the imperative. Rather, we introduced a new imperative by having the target color change. In this way, we separated the CS and the imperative. For the negative control, we extended the interval between the CS and the imperative signal (to ~1,000 ms). With this delay, Pavlovian effects were no longer observed. We also ran an experiment with the new imperative but now with a CS-US interval similar to that used in the original study. Here the Pavlovian effects were again manifest.

These new experiments are presented as Experiments 2 and 3 in the revised manuscript. We thank the reviewers and editor for pushing us to run this study—the direct test of the timing constraint provides an important addition to the argument that adaptation can be conceptualized as a form of conditioning.

Exp 1 probe phase results: Whilst not significant (n-1 x n interaction), there seemed to be a clear difference between CS+ trials within the different trial n-1 contexts (Figure 1D). In fact, this difference seems bigger (and as consistent) as the meaningful/significant differences which are focused on. Interestingly, the RW model predicts a clear n-1 x n interaction however it is not discussed why (Figure 1F). To me it seems that the behaviour (at least partially) and model reflect an interaction between trial n-1 and n during probe trials however this is currently not discussed. Could the authors elaborate on this result and include this in the article?

As noted by the reviewer, in the probe phase of Experiment 1, the Pavlovian effect (the difference in heading angle between CS+ and CS- trials on trial n) seems to depend on the CS presented on the previous trial. However, this interaction is not predicted by the Rescorla Wanger model: It predicts the same Pavlovian effect (difference between CS+ and CS- on trial *n*) regardless of whether trial *n-1* is CS+ or CS- (Author response image 2). While the data in Figure 1F might suggest an interaction, the interaction term is not significant [F(1,15)=2.03, p=0.175,, BF_10_=0.69, η_p_^2^=0.12]. When looking at the individual data (Author response image 2), one can see that they are quite noisy. Importantly, the results from the probe phase of the new Experiment 3 provide a second test since this is essentially a replication of Experiment 1 but with the new imperative. Here, too, the interaction is not significant (Author response image 2, [F(1,63)=0.86, p=0.358, BF_10_=0.19, η_p_^2^=0.01]). In fact, the trend here is in the opposite direction of that observed in Experiment 1.

**Author response image 2. sa2fig2:** Differential conditioning: Rescorla-Wagner predictions and experimental results. (A) Rescorla-Wagner predictions for the change in heading angle during the probe phase in differential conditioning. The model predicts no interaction between the CSs presented on trial n-1 and trial n (the dotted arrows that represent the Pavlovian effect are parallel). (B, C) Experimental results from the probe phase in Experiment 1 (B) and Experiment 3 (C).

We note that based on the Rescorla-Wagner model, the pattern of results across conditions could point to a main effect of trial n-1, but again, this effect did not reach significance. We address this general issue in the revised Results section:

“Despite the absence of feedback in the probe phase, the change in heading angle on trial n (CS+ or CS-) should also depend on the CS presented on trial n-1. After an association is formed, the motor state, namely the actual heading angle, fluctuates as a function of the presented CS, decreasing on CS- trials (due to extinction) and increasing on CS+ trials (due to conditioning). As such, the heading angle on a CS+ trial should increase more if it follows a CS- trial than if it follows a CS+ trial. Similarly, the heading angle on a CS- trial should decrease more if it follows a CS+ trial than if it follows a CS- trial. While the data appeared to follow this pattern, there was no main effect of the trial *n-1* CS {-0.44°, [-1.11° 0.24°]; F(1,15)=1.61, p=0.224, BF_10_=0.56, η_p_^2^=0.10}. This may be because, statistically, the main effect of trial *n*-1 considers consecutive trials with a repeated CS (CS+ that follows CS+, and CS- that follows CS-) where the change in heading angle is minimal. Crucially, the Pavlovian effect (the difference between the change in heading angle on CS+ and CS- trials on trial n) should not depend on the previous trial. Consistent with this prediction, the trial *n-1* CS x trial *n* CS interaction was not significant [F(1,15)=2.03, p=0.175, BF_10_=0.69, η_p_^2^=0.12].”

Magnitude of effect (supplementary S1 figure): Suppl Figure 1 needs to be in the main article as it provides context and some 'raw' data. It would also be beneficial to have in the main article a similar figure but with the CS+ and CS- trial types separated. Suppl Figure 1 is important as it highlights that the conditioning effects are relatively small. This needs to be explained/mentioned in the results i.e., that the differences between CS+ vs CS- (1-degree) are approx. 6% of the total adaptation that occurred (15-degrees). I suppose this is referred to within the final exp 1 analysis (Figure 2B) but its not explicit. Although I believe the results are important, the article currently reads as if these conditioning effects were large when in fact other people might conclude that conditioning had little impact on adaptation (as similar adaptation (approx. 15-degrees) was observed across both contexts (assumed as this is not shown) and performance looks very similar to exp 2 where there was no 0-degree context). There needs to be some acknowledgement of the fact that while these conditioning effects appear meaningful, they were small (with most participants showing less than a 1-degree difference between contexts).

We agree with the reviewer’s assessment and now include Figure S1 in the main article.

When looking at the time course of the heading angle data for CS+ and CS- trials (Author response image 3), the difference between conditions is small. However, we are hesitant to draw strong inferences about the effect size from this particular analysis for two reasons:

1. As mentioned above, the heading angle is influenced by the CS presented on trial n and by the motor state which is affected by the CS presented on trial n-1.

2. Between-participant variability in overall adaptation level somewhat masks the differential conditioning effects. The key tests are the within-participant effects. Thus, we focus our core analyses on trial-by-trial changes in heading angle. We believe this is a direct measure for how contextual cues influence behavior within each participant.

**Author response image 3. sa2fig3:** Time course of heading angle in Experiment 1 on CS+ (light blue) and CS-(dark blue) trials. .

The overall change in heading angle from learning (raw adaptation level) and the trial-by-trial change in heading angle are different measures. As such, we do not think it would be valid to directly compare them to assess the relative effect size. A more valid comparison is to compare the trial-by-trial Pavlovian (trial n) effect to the trial-by-trial Adaptation (trial n-1) effect observed during the acquisition phase. Such a comparison reveals that the Pavlovian effect is quite large. For example, in Experiment 1, the Pavlovian effects in the acquisition (0.82° on average) and probe (1.06°) phases are 55% and 71% of the Adaptation effect (1.50°), respectively. In sum, while we agree that the primary signal that drives the learning curve is the typical previous trial effect, the Pavlovian effect is also profound.

Exp 2 results: It was unclear to me why the RW model would predict a negative heading angle within the single CS conditions (Figure 3C)? I understand that a weaker conditioning response would be expected due to compound conditioning, but would you not expect this still to be positive? Why would the model predict extinction to occur and why is this seen in the behaviour? The details of this result (and the predictions of the model) are currently not discussed in sufficient/any detail.

The prediction regarding the negative heading angle is important and we see that the original submission lacked clarity on this point. The negative heading angle for the single CS conditions in Experiment 2 (now Experiment 4) is expected due to a combination of two factors: (1) We focus on trial-by-trial *changes* in heading angle, manifest as variation around the mean of the *actual* heading angle, and (2) the heading angle will decrease during the probe phase due to extinction given the absence of feedback.

To clarify these points, we added the following:

“We note that the median trial-by-trial change in heading for the tone-alone and light-alone trials is negative. This is because this measure reflects modulations in behavior around the mean heading angle, and this mean decreases over time due to extinction given the absence of the error feedback. This general trend should also vary for the different CSs depending on their relative salience (Equations 3) (Kamin, 1967). Crucially, the additivity principle posits that there should be a negative correlation between the associative strengths of competing CSs (Rescorla and Wagner, 1972). That is, if a strong associative bond is formed between one CS and the US, this will come at the expense of the associative strength accrued by competing CSs (Equation 3) given the capacity limit on associability (*V*). This prediction was strikingly confirmed in an analysis of the heading angle changes on tone-alone and light-alone trials: Participants who were more sensitive to the tone stimulus were less sensitive to the light stimulus, and *vice versa* (Figure 8E, Pearson correlation: r=-0.72, p<0.001, BF_10_=163.2).”

What is the CS (confusion between results and discussion)? Between lines 395-410 the authors describe the primary CS as being the heading angle ('the movement plan itself, rather than the target cue, that constitutes the primary CS'), however in the results (lines 74-96) they describe the CS as being the arbitrary cue ('When considered through the lens of classical conditioning, the arbitrary cues are the conditioned stimuli (CSs)') and the CR being the heading angle ('the conditioned response (CR) would be the movement heading angle elicited by a CS'). As a result of this discrepancy, I found this section of the discussion very confusing (lines 387-410 and then again from line 433). Are the authors saying that sometimes the heading angle/plan is the CS and other times it is the CR…? How does this all align? One can see why a suggestion for a figure showing this mechanism visually is suggested below.

We now see that the original version was confusing on this issue. To form an analogy between sensorimotor adaptation and classical conditioning, we aimed to illustrate that adaptation to sensorimotor perturbations adheres to key rules of associative learning, including associability with arbitrary cues.

We opted to use tone and light cues as arbitrary contextual CS since they are often used in classical conditioning paradigms as CSs (e.g., in eyeblink conditioning). These cues are not present in standard sensorimotor adaptation tasks, making it important to consider what other CSs might be present in standard tasks (assuming one wants to consider such tasks as subject to the constraints of associative conditioning). This led to the proposal that the CS in standard adaptation tasks is the movement plan (defined by the position of the target). The movement plan shows features of a conditioned stimulus (e.g., associability and generalization). Moreover, the onset of the target is a salient cue that usually serves as an imperative for movement initiation, thus complying with the timing requirements needed to form an association.

We expected the movement plan would remain a salient CS in our experiments; indeed, we were concerned that it would overwhelm effects from the tone and/or light CSs. For this reason, we chose to use a single target position, have the target remain on the screen throughout the experiment, and make the light and tone be the imperative signals. In this way, we made these CSs more salient while imposing the timing requirements between these cues and the cursor feedback (and test the latter in new Experiments 2 and 3). We now clarify this point in the *Results* section:

“When considered through the lens of classical conditioning, the target appearance could be viewed as a conditioned stimulus (CS) given that it is presented just before the sensory feedback (the unconditioned stimulus, US) resulting from the movement (Figure 1B). To test the efficacy of arbitrary cues as conditioned stimuli, the onset of the tone or light served as the imperative, with the target visible at its fixed location throughout the experimental session (Figure 1B).“

And return to this issue in the *Discussion*:

“These temporal constraints are likely built into standard adaptation tasks: The target is a salient stimulus that defines the task goal and movement plan, and its onset usually serves as the imperative for movement initiation. RTs in these tasks are typically below 500 ms (Kim et al., 2018, 2019; Avraham et al., 2021). As such, a tight temporal link is established between target appearance and movement, echoing (perhaps inadvertently) the CS-US temporal constraints essential for cerebellar-dependent conditioning. Under these conditions, the target can be viewed as a highly effective contextual cue.”

Link between this work and recent contextual inference model by Heald et al.,: Could the authors provide a more direct comparison between the current work and the contextual inference model by Heald et al., (in discussion)? The authors currently say 'their model suggests that spontaneous recovery and consistency effects emerge due to contextual inferences that likely interact with deliberate changes in explicit strategies' however does this align with the current work? How does contextual inference align/differ with these conditioning mechanisms? In future work, the authors seem to want to explain the phenomena (spontaneous recovery etc) recently explained by this model and I am interested to know whether these are competing explanations or are explaining the same mechanism or are different but complementary?

This is a great suggestion. We certainly have given considerable thought to how the ideas presented in this paper relate to the contextual inference (COIN) model proposed by Heald and colleagues. The two share a core premise, namely that context has a key role in sensorimotor learning. The Rescorla-Wagner model describes, in a simple way, how associations between contextual cues and sensory outcomes allow the expressions of context-dependent motor memories. This idea is central to the COIN model. One difference between the approaches, however, is in how context is inferred. In the COIN model, context is probabilistic, and it is *inferred* by the learner given their experience with sensorimotor feedback and environmental cues. In our application of RW to adaptation, the context is deterministic – it is specified by the sensory cue presented on a given trial.

While both models offer accounts for how context or environmental cues could separate memories, they likely differ in terms of the neural systems involved and how they account for some ubiquitous phenomena described in the sensorimotor learning literature (e.g., spontaneous recovery, consistency effects, savings). One complication here stems from the fact that sensorimotor learning, including simple adaptation tasks, entails the operation of multiple learning processes. One distinction is between implicit, automatic processes that maintain the calibration of the sensorimotor map and processes that facilitate learning by changing action plans (e.g., re-aiming), including those that may be volitional and under explicit control. For example, while there are studies claiming that consistency effects and savings are due to implicit changes in sensitivity to motor errors (Herzfeld et al., 2014, Coltman et al., 2019, Albert et al., 2021), a growing body of work suggests that these phenomena primarily reflect the use of explicit strategies (Morehead et al., 2015, Haith et al., 2015, Leow et al., 2019, Schween et al., 2019, Avraham et al., 2020, Avraham et al., 2021).

In our view, the contextual inference process suggested by Heald and colleagues is most likely associated with the explicit operations that lead to modifications in action plans, and their acquisition is perhaps dependent on the medial temporal lobe memory system (Collins and McDougle 2021; McDougle et al., 2022). By contrast, we have used the clamp method in our studies to limit learning to the implicit adaptation system, asking if and under what constraints is this system sensitive to contextual cues. By adopting a conditioning perspective, we find that some of these phenomena emerge from the relatively simple learning rules of the Rescorla-Wagner model.

While the specific mechanisms underlying consistency effects and savings in motor adaptation are debatable, we note that we are not arguing that the implicit motor system is *not* involved at all in higher-order effects like spontaneous recovery or interference. Indeed, in a recent study we have reported that in a standard savings design of visuomotor adaptation, the implicit system is actually attenuated upon relearning (Avraham et al., 2021, *PloS Biol*). Interestingly, such attenuated relearning has been reported in other conditioning tasks (e.g., fear conditioning, see Bouton 1986). This attenuation appears to be driven by an interference effect that arises from the feedback experienced during the extinction/unlearning phase after the first phase of adaptation. If feedback is removed during this phase, the attenuation effect is eliminated (Avraham and Ivry, *in preparation*). Along the same lines, spontaneous recovery in fear conditioning is also sensitive to the procedure of extinction and to the context in which it is experienced (Bouton 2002).

We recognize that the Rescorla-Wagner model, at least in its basic form, is unable to capture various higher–order learning effects. Recent advances in the reinforcement learning literature have shown that the combination of a Rescorla-Wagner like model with a Bayesian inference process, provides one solution. Drawing inspiration from this literature, we believe our study offers an exciting step towards the evolution of a similar integrative approach. Nonetheless, we do not directly address things like spontaneous recovery and savings in the current paper, leaving these issues for future studies.

We have revised the *Discussion* to address these issues.

“The Rescorla-Wagner model offers an account of how arbitrary sensory cues, visual targets, and movement plans can serve as CSs for adaptation. There are, however, some notable phenomena in the sensorimotor adaptation literature that are not accounted for by our application of the Rescorla-Wagner model, nor by the basic state-space model. Given the prominent role of the state-space model in the sensorimotor adaptation literature, variants of this model were developed to capture these observations. One such effect is spontaneous recovery, the re-manifestation of a previously adapted state in the absence of error feedback. A variant of the state-space model, one that allows for multiple states with different learning and forgetting rates, can capture spontaneous recovery (Smith et al., 2006). A second example is the relationship between learning rate and environmental consistency; learning is faster in response to a consistent versus inconsistent perturbation (Albert et al., 2021; Avraham et al., 2020; Gonzalez Castro et al., 2014; Herzfeld et al., 2014; Hutter and Taylor, 2018). To account for this effect, the state-space model can be modified to allow the learning rate to vary with experience (Herzfeld et al., 2014). A third example is savings upon relearning, where faster adaptation is observed upon a second exposure to the same perturbation (Huang et al., 2011; Krakauer et al., 2005). Studies have modeled savings using multi-rate state-space models (Smith et al., 2006), learning rate modulation (Herzfeld et al., 2014), or a combination of these factors (Zarahn et al., 2008; Mawase et al., 2014).

These higher-order phenomena have also been addressed in a new model of sensorimotor learning, one that takes a Bayesian inference approach (Heald et al., 2021). The contextual inference (COIN) model posits that motor adaptation arises from two interacting mechanisms: Proper learning, which involves the creation and updating of context-specific memories, and apparent learning, a process of inferring the current context to determine which memory to express (recall). Unlike conventional context-independent models of motor adaptation, the COIN model stresses the importance of the environment in guiding memory retrieval. In one sense, our implementation of the Rescorla-Wagner model follows a similar philosophy. However, it maps to the proper learning component of the COIN model since we assume that the current context is determined by the actual properties of the environment in an all-or-none manner (e.g., the CS presented by the experimenter). The COIN model suggests that spontaneous recovery, consistency effects and savings may emerge due to contextual inferences; these inferences likely involve cognitive systems related to executive function and nonmotor memory systems (Collins and McDougle, 2021). Consistent with this view, growing evidence show that these phenomena are, in large degree, the result of strategic changes (Haith et al., 2015; Morehead et al., 2015; Leow et al., 2020; Avraham et al., 2020, 2021; Wang et al., 2022).

While the basic Rescorla-Wagner model is unable to account for higher order effects such as spontaneous recovery or savings, we envision that complimenting it with more sophisticated, Bayesian inference models could readily accommodate these phenomena. In fact, models that combine associative mechanisms with some form of a Bayesian inference process that carves the world into distinct contexts have successfully captured a range of complex phenomena in classical conditioning and reinforcement learning (Collins and Frank, 2013; Courville et al., 2006; Gershman, 2015; Kruschke, 2008). We believe that our results lay the foundation for adopting a similar approach to study implicit sensorimotor adaptation and perhaps a more general account that captures the operation and interaction of multiple learning processes.”

Use of supplementary figures: Why did the authors decide to put so much of the important detail into the supplementary? In my opinion, all of this should be in the main article.

We have pivoted and now include all of the supplementary figures in main text.

Normality of data and presentation of individual data: There is no mention of any assessment of data normality, was the data normally distributed? In addition, a greater amount of individual data should be shown rather than mean +- SEM.

We thank the reviewer for flagging the paper on this point. We now mention how normality was assessed and the statistical tests we used in conditions in which there were violations of normality assumptions:

“We followed the assumption of normality when the sample size was larger than N=30 based on the central limit theorem or when the Lilliefors test indicated normality (Lilliefors, 1967). When these conditions were met, we used parametric statistical tests. In all other cases we used non-parametric statistics (see Experiment 4). Family-wise errors in pairwise comparisons were corrected using the Bonferroni correction.”

We have added individuals’ data points for all the presented results.

Figure to represent conditioning mechanism: I would find it very helpful if an additional figure was added which showed the US, UR, CS and CR visually. I kept forgetting how each of them were proposed to be represented in the task.

We now include figures that illustrate the elements of classical conditioning in the context of our tasks (Figures 1B, 6A, 7A).